# Multimodal deep learning using on-chip diffractive optics with in situ training capability

Junwei Cheng[1], Chaoran Huang [2], Jialong Zhang[1], Bo Wu[1], Wenkai Zhang[1], Xinyu Liu[1], Jiahui Zhang[1], Yiyi Tang[1], Hailong Zhou[1], Qiming Zhang[3], Min Gu[3], Jianji Dong[1,4] ✉ & Xinliang Zhang[1,4]

Multimodal deep learning plays a pivotal role in supporting the processing and learning of diverse data types within the realm of artificial intelligence generated content (AIGC). However, most photonic neuromorphic processors for deep learning can only handle a single data modality (either vision or audio) due to the lack of abundant parameter training in optical domain. Here, we propose and demonstrate a trainable diffractive optical neural network (TDONN) chip based on on-chip diffractive optics with massive tunable elements to address these constraints. The TDONN chip includes one input layer, five hidden layers, and one output layer, and only one forward propagation is required to obtain the inference results without frequent optical-electrical conversion. The customized stochastic gradient descent algorithm and the drop-out mechanism are developed for photonic neurons to realize in situ training and fast convergence in the optical domain. The TDONN chip achieves a potential throughput of 217.6 tera-operations per second (TOPS) with high computing density (447.7 TOPS/mm$^2$), high system-level energy efficiency (7.28 TOPS/W), and low optical latency (30.2 ps). The TDONN chip has successfully implemented four-class classification in different modalities (vision, audio, and touch) and achieve 85.7% accuracy on multimodal test sets. Our work opens up a new avenue for multimodal deep learning with integrated photonic processors, providing a potential solution for low-power AI large models using photonic technology.

Multimodal deep learning is a novel deep learning technique that can process and analyze multiple types of data. With the explosive growth of the artificial intelligence generated content (AIGC) in the past few years, multimodal model is an emerging trend in the field of large language models[1–4]. Thanks to multimodal technology, large artificial intelligence (AI) models represented by ChatGPT can break through the boundaries of single modality. At present, the latest version of GPT-4 has successfully realized the processing of multimodal data such as images and audio. However, large AI models require huge computing resources and throughput for parameter optimization and inference. The performance of microelectronic processors based on digital-clock platform is limited by Moore's Law[5], and it is difficult to further improve the throughput and energy efficiency to meet the increasing expansion of AI computing overhead. Optical neural networks (ONNs)

[1]Wuhan National Laboratory for Optoelectronics, Huazhong University of Science and Technology, Wuhan 430074, China. [2]Department of Electronic Engineering, The Chinese University of Hong Kong, Hong Kong 999077, China. [3]Institute of Photonic Chips, University of Shanghai for Science and Technology, Shanghai 200093, China. [4]Optics Valley Laboratory, Wuhan 430074, China. ✉e-mail: jjdong@mail.hust.edu.cn

encode information in multiple dimensions of light and implement computation by the propagation of optical signals through photonic devices[6–11]. The concept of propagation-as-computation makes ONNs naturally have high parallelism and high energy efficiency, and it is expected to be a potential solution to low-power AI.

In recent years, a variety of ONN architectures have been proposed and successfully demonstrated in AI tasks, including coherent architectures based on integrated Mach-Zehnder interferometer (MZI) grid[12–19], microring modulator array for wavelength division multiplexing (WDM) processing[20–32], and phase change material (PCM)-based crossbar waveguide routing networks[33–36]. The latest ONN architectures have made great progress with excellent performance, such as throughout up to 11 trillion operations per second[37], compute density over 1 trillion operations per second per square millimeter[38], and are already competitive with digital computers in some classical dataset tasks. However, since the footprint of photonic device units is difficult to shrink[39], the hardware integration remains limited. The computational scale of these architectures is typically limited to 4 × 4 matrix-vector multiplications or smaller[16–18,23,27,40]. Due to the small size of the photonic computing cores, a single inference task requires frequent calls to the photonic computing cores, which leads to frequent optical-electrical conversion and additional energy consumption. As a result, most existing ONN researches still struggle with classic tasks and small datasets, such as MNIST[41] and Fashion-MNIST[42] datasets.

To enable large-scale on-chip ONN architecture, the concept of on-chip diffractive optical neural network (DONN) has recently been proposed[43–45]. The on-chip DONN architecture utilizes a series of silicon slots filled with silica to implement the function of the hidden layers of the neural network, where silicon slots filled with silica act as neurons. Since the geometric dimensions of the silicon slots are designed by numerical simulation, corresponding to the weight parameters of the neurons, the training process is in fact implemented on digital computers rather than in optical domain. Once the photonic chip is fabricated, on-chip diffractive units cannot be tuned as the user expected, thus the weight parameters and functions of the on-chip DONN are also fixed, resulting in the lack of reconfigurability at present. Although on-chip DONN can perform single-modal tasks well, it is difficult to adapt to new types of tasks due to the lack of in situ training of the neural network. A robust model typically requires persistent updates and iterations given the current prevalence of multimodal learning techniques.

Here, we propose a trainable diffractive optical neural network (TDONN) architecture to process and classify multimodal data by light propagation. Leveraging on the superposition and coherence properties of light, massive neurons in the hidden layers can be naturally connected by diffraction in different modal settings. To demonstrate the multimodal capabilities of the proposed architecture, we fabricate a 5-layer TDONN chip on silicon-on-insulator (SOI) platform, and carry out four-class classification experiments in three different modalities, including visual, audio, and tactile data. In the experiments, the tailored stochastic gradient descent algorithm is deployed for the TDONN chip to realize in situ training. The drop-out mechanism is designed for massive trainable neurons to accelerate the convergence speed (36.5% convergence acceleration compared to no drop-out) in the optical domain. After training, the TDONN chip has successfully classified different modal data and achieve 85.7% accuracy on multimodal test sets. Our TDONN chip achieves a potential throughput of 217.6 tera-operations per second (TOPS) with high computing density (447.7 TOPS/mm²), high energy efficiency (7.28 TOPS/W), and low optical latency (30.2 ps). In addition, the standard complementary metal oxide semiconductor (CMOS) process enables low-cost fabrication in commercial foundries without high-resolution lithography. Our work provides a promising high-performance computing hardware for multimodal deep learning and paves the way for large-scale photonic AI models.

## Results
### Design and fabrication of the TDONN chip
Neuromorphic photonics aims to map complex physical models into abstract models of neural networks. Figure 1a illustrates the optical neural network model for the multimodal classification task, and it consists of three parts: an input layer, five hidden layers, and an output layer. After feature extraction and feature fusion, a feature vector is obtained from the datasets of different modalities such as vision, audio and touch, which is used as the input of the neural network. The size of

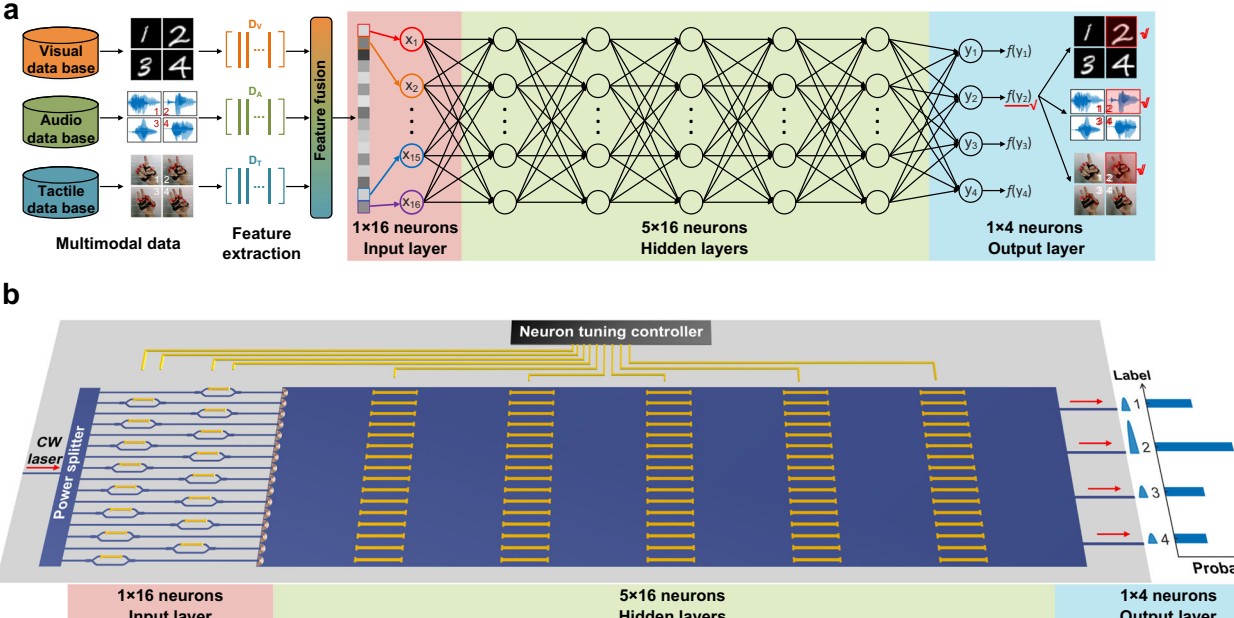

**Fig. 1 | Schematic of on-chip diffractive optics for multimodal deep learning. a** Optical neural network model for the multimodal classification task. **b** The TDONN chip contains input, hidden and output layers, and supports in situ training and inference in optical domain.

the feature vector matches the number of neurons in the input layer, and each vector element is encoded into the optical signal by means of intensity modulation. In the hidden layers, the neurons are arranged according to a multi-layer layout, and the connection weights between each neuron are adjusted during training to achieve the target function. Therefore, trainable neurons are an important prerequisite for reconfigurable TDONN. The data flowing into the output layer are considered as the output vector, and the elements of each output vector correspond to output channels. The output power of each channel is mapped to the probability in the inference result. By comparing the output power of each port, the label with the highest probability can be obtained as the inference result of the TDONN architecture. A comparison diagram between TDONN and the traditional matrix multiplication architecture is provided in Supplementary Note 1.

We design and fabricate a TDONN chip to demonstrate the proposed architecture. The device is fabricated on an SOI wafer with 220 nm top silicon and 2 μm thick buried oxide substrate. Figure 1b depicts the structure of TDONN chip with input, hidden and output layers corresponding to the neural network model in Fig. 1a. The input layer consists of 16 intensity modulation units, which are used to load the input data, and the element values of the feature vector are encoded into the intensity of the optical signal through the on-chip modulation. The hidden layers consist of tunable diffractive units arranged in five layers with 16 tunable diffractive units per layer. These diffractive units are used to simulate the neurons in hidden layers. Without loss of generality, a metal micro-heater array is used to adjust the parameters of the diffractive units. Each micro-heater can be regarded as an independent trainable diffractive unit. By adjusting the voltage applied to each heater, the local temperature in the vicinity of each independent diffractive unit can be precisely controlled, thereby change the local effective refractive index in the corresponding region in the hidden layer. The output layer includes four output ports, and the output optical signal is received by a 4-channel photodetector (PD) array. The probability distribution of each label can be obtained by detecting the output optical power in real time by the PD array. The weight parameters and function of the TDONN chip can be configured according to specific multimodal task through in situ training. When the training is complete, only one forward propagation of light is needed to realize the processing and computation of multimodal information such as vision, audio, and touch. The electromagnetic propagation of optical signals in TDONN chip can be regarded as the repetition of two-dimensional free-slab diffraction and phase modulation implemented by phase shifter array. The complex transmission matrix of free-slab diffraction can be described using the Huygens–Fresnel principle:[46]

$$w = \frac{1}{j\lambda} \frac{1 + \cos(\boldsymbol{n}, \boldsymbol{r})}{2r} e^{ikr} \tag{1}$$

where $w$ is the complex transmission coefficient between the two points on the input and output plane. $\lambda$ is wavelength of input light. $\boldsymbol{n}$ is the normal vector of input plane. $\boldsymbol{r}$ is a vector formed by the two points. $k$ is wavevector of light in the slab waveguide. To help understand the physics of diffraction in the TDONN architecture, we build a general theoretical model of diffractive neural networks to describe the light forward propagation and the error backward propagation, and the details can be found in Supplementary Note 2 and Supplementary Note 3. Massive trainable parameters are fundamental to ensure the reconfigurability and multimodal capability of the TDONN. The slab region with phase shifter array constitutes a tunable matrix containing 16 degrees of freedom, and a highly intricate linear matrix with strong reconfigurability can be obtained by cascading the free-slab diffraction and tunable phase modulation matrix. In actual experiments, the waveguide boundary will reflect some light, which

may reach the next diffractive layer, thus the training of photonic chip does not strictly follow the theoretical model. To effectively optimize the diffraction network in experiments, we treat TDONN as a 'black box' with numerous trainable weight parameters, and update the weight parameters of the diffractive network by detecting the optical response of the output layer in real time.

Figure 2a, b shows microscope images of the input and hidden layers of the TDONN chip, including the 16 intensity modulation units and 80 diffractive units integrated on the multi-port silicon slab waveguide. The intensity modulation unit is designed as MZI structure to load the input data, and a thermal isolation slot is etched between adjacent intensity modulation units to reduce thermal crosstalk. Each diffractive unit corresponds to a TiN heater, which is 100 μm long and 3 μm wide, and the spacing between the centers of adjacent heaters is 20 μm. These heaters are placed in five layers with a period of 280 μm in a 1.35 mm × 0.36 mm area. The period of hidden layers is determined by numerical simulation and the details are provided in Supplementary Note 4. Figure 2c shows the packaged chip, where vertical grating coupling and wire bonding have been completed, and optical and electrical I/Os are used for light propagation and the parameter control of diffractive units, respectively. The optical signal encoded with the multimodal data is injected into and outputs from the TDONN chip through a vertical grating coupler, and the programmed voltage control is applied onto the on-chip heater through wire connection. Phase shift is introduced by applying voltages on resistive heaters to effectively control the light propagation in TDONN (see Supplementary Note 5 for detailed information). The TEC module is mounted below the TDONN chip to compensate for temperature variations. To achieve high-precision voltage scanning, we specifically develop a programmable power supply with 16-bit resolution to precisely regulate the intensity modulation unit and the diffraction unit. As shown in Fig. 2d, the experimental setup includes photonic and electronic components, and the integrated photonic and control components can work together under a single field programmable gate array (FPGA) control framework without human intervention. We developed a photonic-electronic prototype based on the TDONN chip, which is shown in Fig. 2e. The prototype can run independently to realize the demonstration function of various modal data. We also developed a graphical user interface (GUI) to display the real-time state of optical training and optical inference with visual, audio, and tactile datasets (see Supplementary Movie 1, 2, and 3, respectively). Figure 2f shows the experimental devices inside the prototype, which operates without benchtop instruments (see Supplementary Note 6 for detailed information).

## Training the TDONN chip

Strong reconfigurable performance is an essential and important capability of computing hardware for multimodal deep learning. The previous on-chip diffractive networks are most likely not reconfigurable. Once the fabrication is completed, its function cannot change, so it can only handle data of a specific modal state. Our TDONN chip can support in situ training and computing in optical domain, and the four-class classification of visual, audio, and tactile data can be realized. The training of TDONN chip is divided into two steps: the first step is to preprocess the input data of different modal states to extract the features, and the second step is to train the tunable diffractive units of the chip to achieve the target function. To train the tunable diffractive units of the chip and achieve the target function, customized gradient descent algorithm and drop-out mechanism of optical neurons are developed and applied in the second step.

The MNIST dataset[41] is used for the visual dataset and the Spoken_numbers_pcm dataset[47] is chosen for the audio dataset. For the tactile dataset, we use commercial somatosensory gloves for data acquisition since there is no suitable publicly available dataset. Five sensors are set on the somatosensory glove to detect the degree of bending of the five fingers, and the intensity of the probe signal is

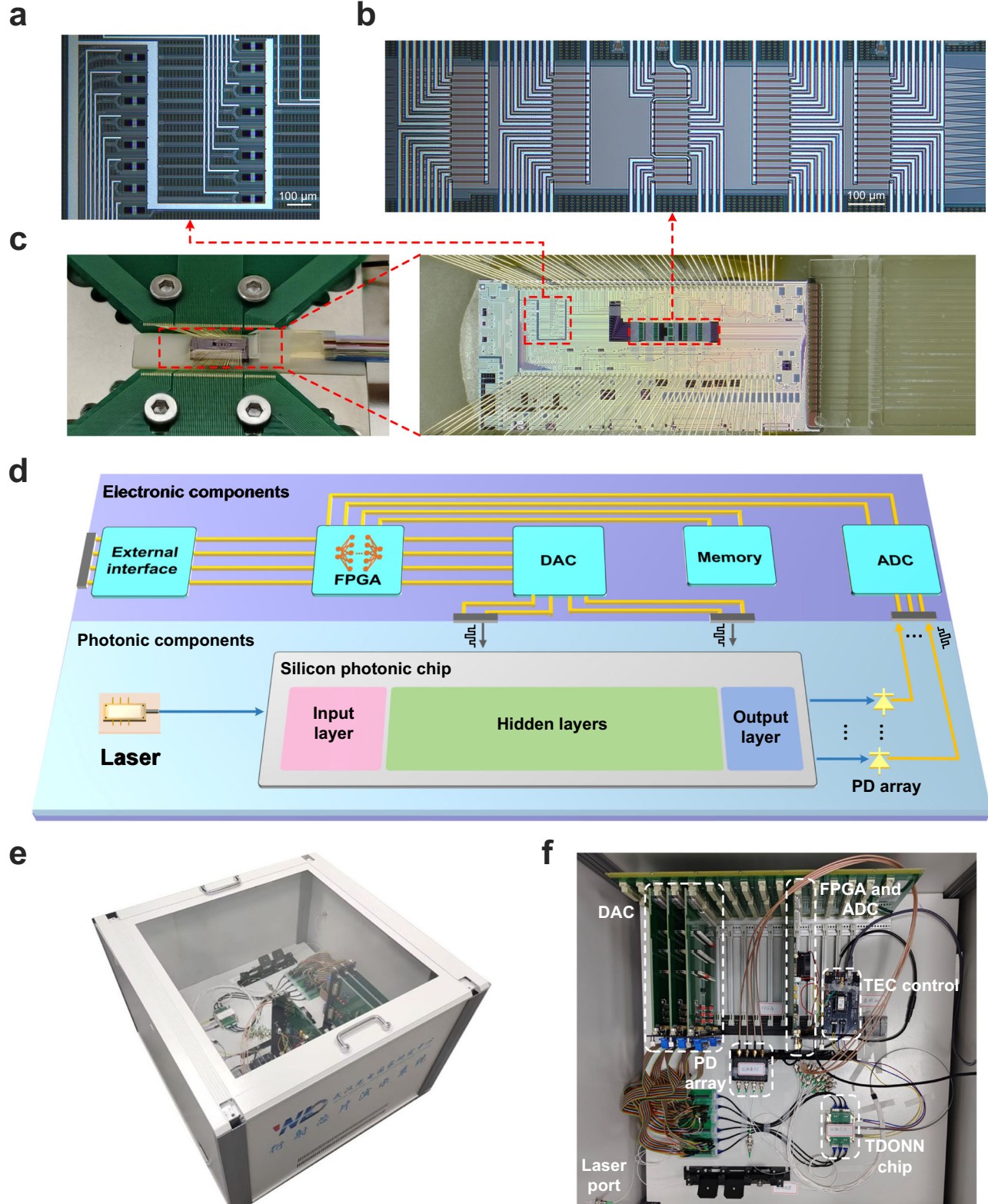

**Fig. 2 | Detailed design of the TDONN chip and the experimental setup. a** The micrograph of the input layer, including 16 intensity modulation units. **b** The micrograph of hidden layers, including 80 diffractive units arranged in 5 layers. **c** The packaged chip. The TDONN chip is wire bonded with a tailored printed circuit board (PCB) and mounted on a thermo-electric cooler (TEC). The optical input and output (I/O) are through the fiber V groove on the top right. **d** The experimental setup contains photonic and electronic components, which can operate without human intervention. **e** A demonstration prototype based on the TDONN chip. **f** Experimental setup in a single FPGA framework without benchtop instruments. DAC digital-to-analog converter, ADC analog-to-digital converter, FPGA field programmable gate array.

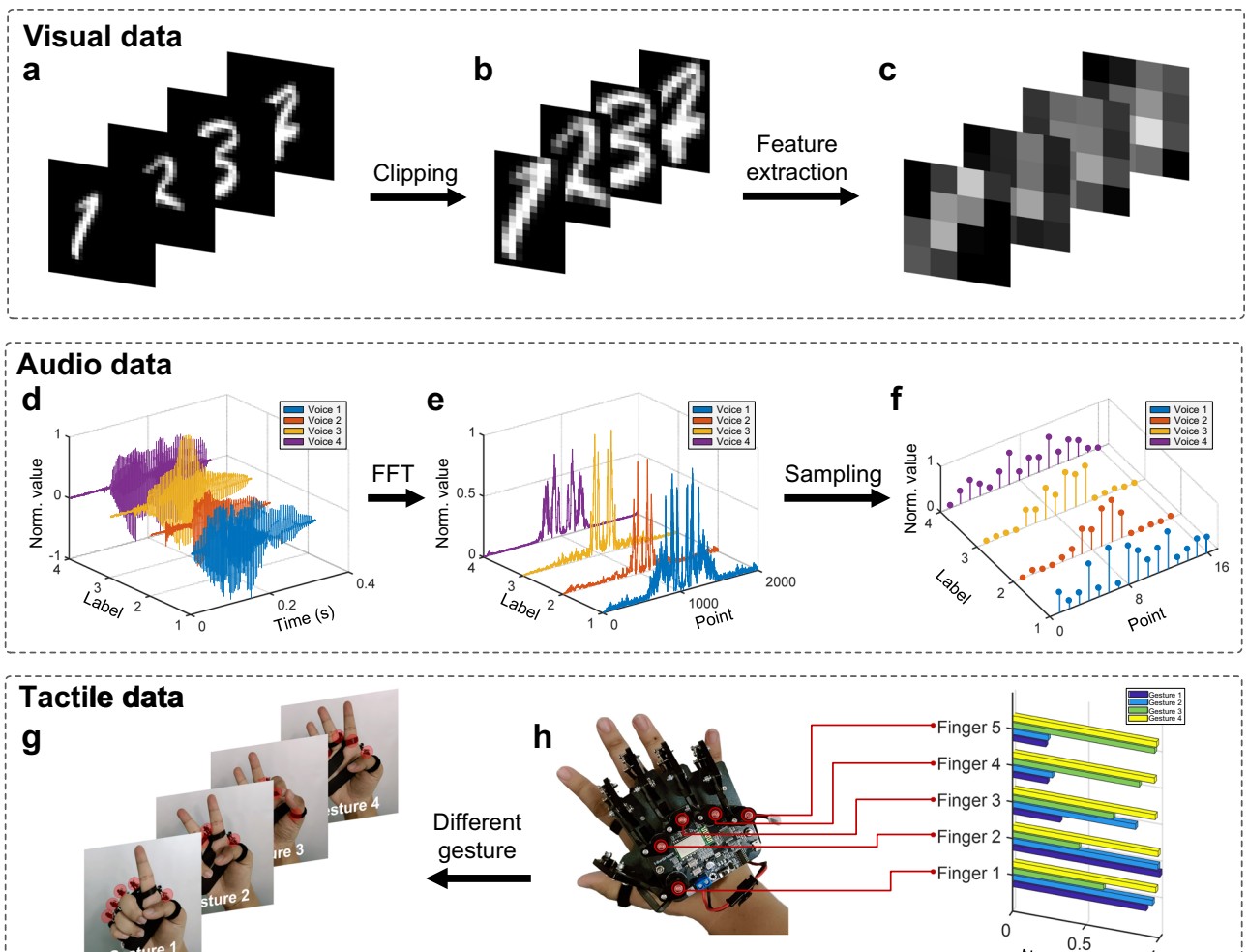

**Fig. 3 | Feature extraction process for multimodal data. a–c** The original image is first clipped and then feature extraction is performed by a fully connected neural network. **d–f** The original audio data is first processed by the 2048-point FFT, and then 16 frequency points are sampled to obtain the corresponding intensity. **g** Five sensors on the somatosensory glove are designed to detect the bending of each of the five fingers when displaying gestures 1–4. **h** Tactile data is collected through a somatosensory glove to obtain the intensity corresponding to each finger.

collected as the tactile data. Considering the scale of audio dataset and tactile dataset, we take 500 groups of data from visual, audio and tactile datasets respectively for training and test, including 1500 groups of data in total. Each modal dataset is divided into training set and test set (train:test=4:1), and set with the same four labels, including one, two, three, and four. In the first step, we specifically design different preprocessing methods for the three modal data to improve the computing efficiency. Figure 3a–c illustrates the preprocessing of the visual data. To eliminate invaluable information, the handwritten digit image is first clipped to retain only the valid area, and then 16 feature values are extracted by a lightweight fully connected neural network. Figure 3d–f illustrates the preprocessing of audio signals. The input signal is a continuous signal in time domain, which needs to be discretized. Through 2048-point fast Fourier transform (FFT) and sampling, the intensity information at 16 characteristic frequencies can be obtained. Figure 3g shows four gestures including one, two, three, and four. Figure 3h shows the somatosensory glove and the mapping between each sensor and the finger. The locations of the sensors have been marked in the figure with red circles.

After preprocessing, each multimodal input is transformed into 16 feature values, which can be encoded on the optical signal by 16 intensity modulation units in the input layer of TDONN chip. Next is to realize the four-class classification task of multimodal data by training

the on-chip diffractive units in the hidden layer of TDONN. Figure 4a shows the flow diagram of the in situ training algorithm, and a cost function (CF) needs to be defined as a figure of merit (FOM) to evaluate the convergence of the training before the algorithm starts its iteration. As shown in Fig. 4a, we define the CF and train the TDONN until its convergence. The CF is defined as

$$CF = \prod_{i=1}^{4} \frac{|M_i \cdot M_{\exp_i}|}{||M_i|| \cdot ||M_{\exp_i}||} \qquad (2)$$

where $M_i$ is the target vector and $M \exp_i$ is the experiment vector consisting of four output ports' real-time optical power during iteration. To reduce the complexity of training, the drop-out check process is introduced. If the heater shows no effect on the improvement of the CF after $N$ iterations, it will be excluded in the following iteration process. Here, we use the training process of the image classification task as an example. The CF gradually increases over iterations and converges after 1500 epochs shown in Fig. 4b. After training 500 epochs, the normalized CF achieves 0.8, which indicates the possibility of the predicted label is apparently higher than the other three labels. The subsequent iterations aim to further enlarge the contrast thereby obtaining the satisfactory classification results. Although crosstalk may exist in the training process of 80 heaters, it

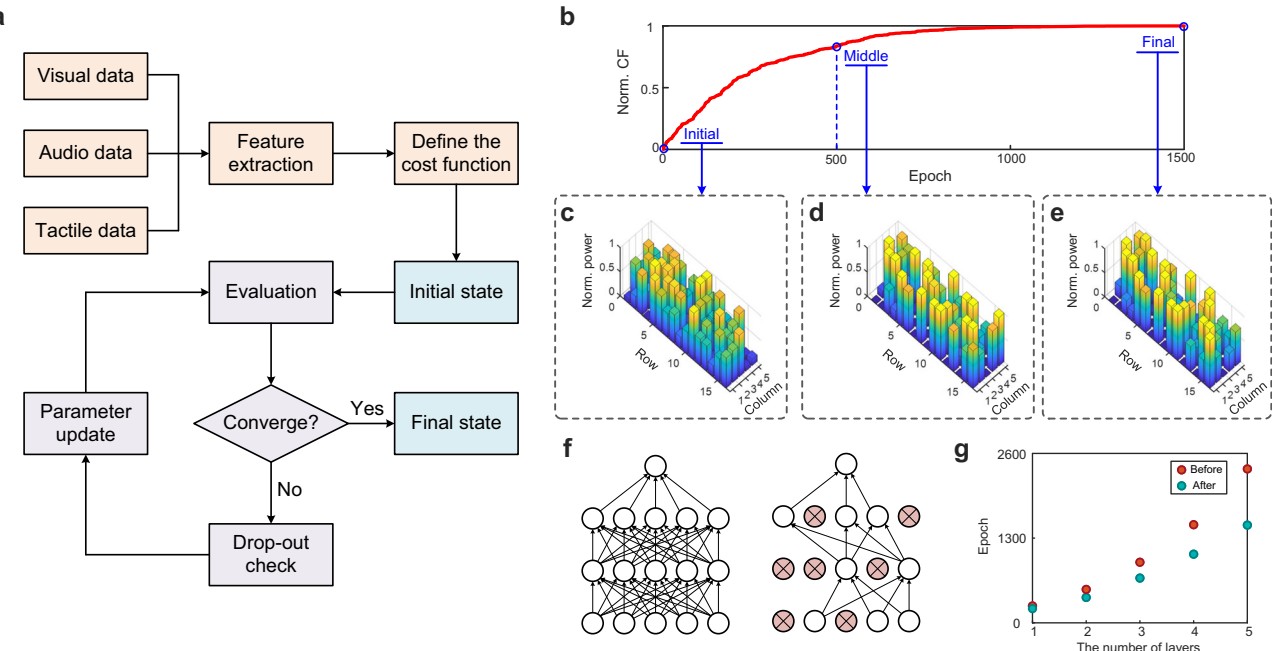

**Fig. 4 | Training of the TDONN chip. a** The flow diagram of the in situ training process. **b** The cost function versus training iterations in the image classification task. **c**–**e** The normalized power distributions of on-chip diffractive units. **f** Conceptual diagram of the drop-out algorithm. **g** The difference in the number of iterations before and after deploying the drop-out algorithm with the different number of layers.

does not affect the training result, because the overall parameters are jointly configured. Once training is over, the TDONN can execute the task at the speed of light. And it can switch to another task via reconfiguring the voltages applied to the heaters. In the training of electrical neural networks, drop-out algorithm is widely used to enhance the generalization ability of the model. Specifically, the neuron will stop working with a certain probability when the data is propagated forward. Inspired by this idea, we developed an optical drop-out algorithm for the in situ training of photonic neurons to accelerate the convergence of TDONN. Figure 4f shows the schematic diagram of the drop-out algorithm. Firstly, an iteration threshold $T_{iter}$ is set for each neuron in the hidden layer of TDONN, and $T_{iter} = 10$ in the experiments. During the iteration process, if the neuron still cannot increase CF after $T$ adjustments, the neuron is set to be inactivated, and in the following iterations, the inactivated neuron will not be adjusted. As the training progresses, the number of deactivated neurons gradually increases, and only the remaining activated neurons need to be tuned, which significantly reduces the workload. In addition, we test the effect of the drop-out algorithm deployed in TDONN with different number of layers, and their required number of iterations for the same task before and after deploying the drop-out algorithm is shown in Fig. 4g. It can be observed that when the number of layers of TDONN is less than 3, the acceleration convergence effect of the drop-out algorithm is not obvious, but the acceleration will become more obvious with the increase of the number of layers, and the acceleration can reach 36.5% when the number of layers is 5.

## Multimodal inference implemented by the TDONN chip

In this section, we use the datasets of three different modalities to verify the multimodal processing capability of the TDONN chip. The TDONN chip includes one input layer (16 neurons), five hidden layers (80 neurons), and one output layer (4 neurons), for a total of 100 on-chip neurons. After preprocessing the multimodal input data, 16 features of image, audio and tactile information can be obtained. The 16 input features are loaded onto the 16 intensity modulation units and encoded on the optical signal in the input layer. The optical signal

propagates directly into the slab waveguide, which then propagates 170 μm through the slab waveguide to reach the first hidden layer and passes through the remaining hidden layers in turn. After the light is emitted from the last hidden layer, it propagates 300 μm to reach the output layer. The output layer has four output ports correspond four detector regions in a linear arrangement. Since each detection region is assigned a specific category, this TDONN chip can achieve a four-class classification task. To equip the TDONN chip with the ability to handle multimodal classification tasks, we first need to adjust the state of each on-chip diffractive units in the hidden layer of TDONN. After that, the chip can directly obtain the inference result based on the power distribution of the output ports with only one forward propagation of light. Figures 5a–d, e–h, i–l shows the probability distribution of classification on one set of visual, audio, and tactile modality respectively, where the peak probability dominates in each case. The experimental results show that the TDONN chip has the ability to perform multimodal classification tasks in the optical domain by in situ training in the optical domain. To further validate the universality of the inference performance of the TDONN chip, 100 test data in each modality are examined by the TDONN chip. Figure 5m–o depict the confusion matrices of test data for three modalities, where the accuracy of classification can reach 86%, 82% and 89%, respectively. And the average accuracy is 85.7%. To better demonstrate the TDONN chip's capability, we further use the TDONN chip to classify more gestures, and the results are provided in Supplementary Note 7. Results show that three different modal datasets with four classes in each modality are successfully classified by the TDONN chip with performance comparable to the digital computer. The classification accuracy can be further improved by the expansion of input dimension and depth of the network.

## Discussion

Our TDONN chip has successfully implemented four-class classification tasks based on datasets in these three modal types. The TDONN architecture incorporates the idea of in situ training, which can use light propagation and real-time feedback of optical response

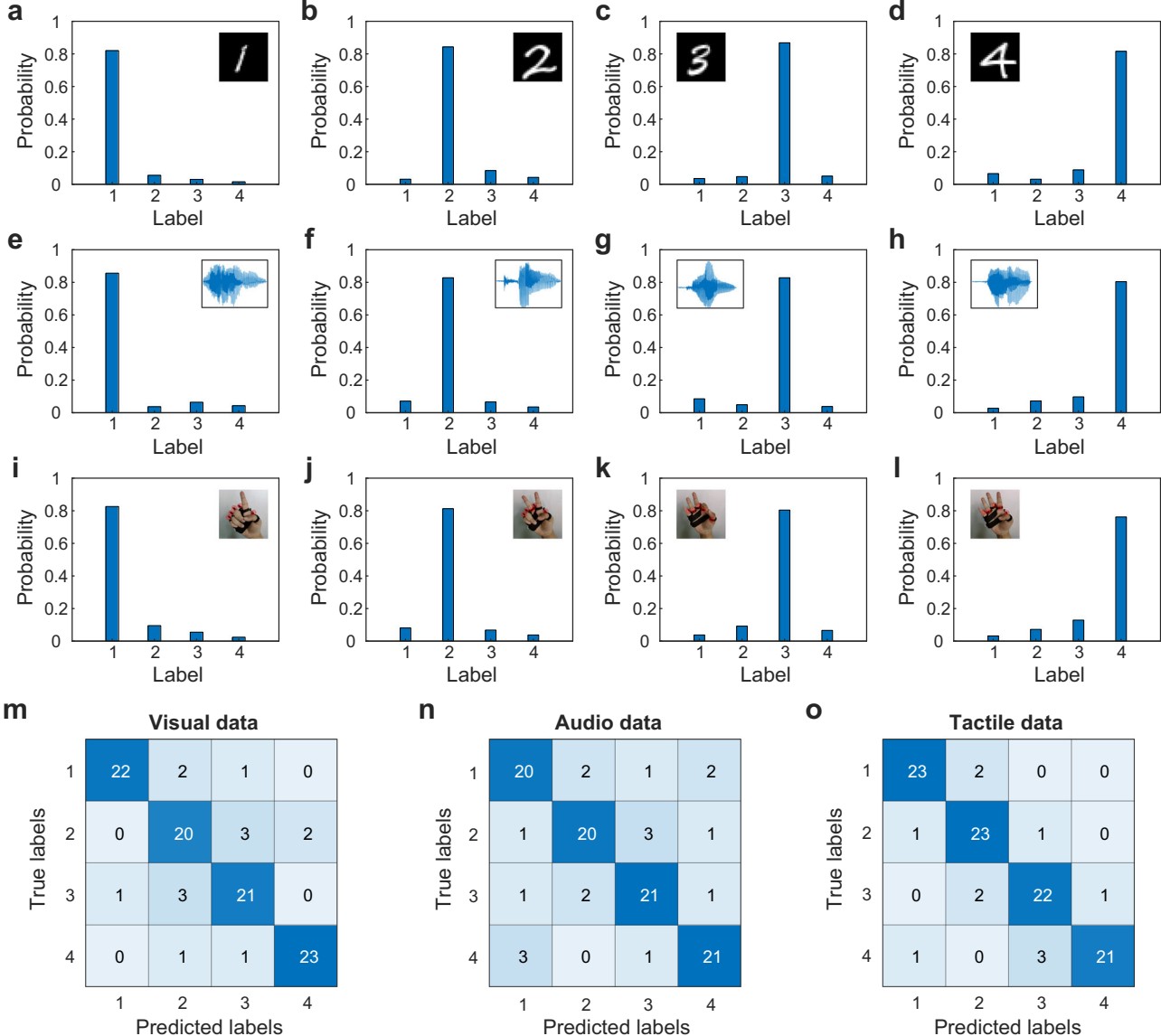

**Fig. 5 | Inference results of multimodal data.** Inference probability distribution of **a**–**d** visual information, **e**–**h** audio information and **i**–**l** tactile information. Confusion matrix of 100 test data for **m** visual information, **n** audio information and **o** tactile information.

to realize training and inference directly in the optical domain, thereby avoid computationally intensive electromagnetic simulation. More importantly, due to the inevitable errors caused by the fabrication process, the mainstream DONN chips need to pre-design external compensation algorithms in digital computers, and the output of the photonic chip needs to be converted into electrical signals for post-processing before the final inference result can be obtained. In contrast, the output of our TDONN chip produces an optical inference result without deploying external compensation algorithms in the digital backend, thus getting closer to an end-to-end ONN architecture, thus reducing energy consumption and latency. The latency of the TDONN chip is approximately 30.2 ps (see Supplementary Note 8 for detailed information). To validate the temperature stability of the TDONN, an experiment is performed with and without using the proposed drop-out method, and the results are provided in Supplementary Note 9. In addition, the results of long-term stability test of the TDONN are provided in Supplementary Note 10. Thanks to the in situ training algorithm specifically designed for the TDONN chip, our photonic-electronic system can realize the cooperative work of integrated photonic devices and

electronic control components in a single FPGA framework without human intervention.

Multimodal deep learning imposes stringent requirements on the throughput, computing density and energy efficiency of computing hardware. Our on-chip TDONN architecture can process multimodal data at the speed of light and with low-power consumption. After the parameters of diffractive units are trained in the optical domain, only one optical forward propagation is needed to directly obtain the optical inference results. In addition, the TDONN architecture is highly scalable, the size of input layer, hidden layer and output layer and the number of hidden layers can be flexibly extended according to the requirements of different multimodal tasks, and its throughput can be significantly improved by using smaller size-tunable diffractive units and arranging with a more compact layout. Suppose that the TDONN has $m$ hidden layers, and $N$ neurons in each hidden layer. According to the estimation method in refs. 13,44, the throughput at a typical 100 GHz detection rate can be estimated as

$$T = 2m \times N^2 \times r\,\mathrm{OPS} \tag{3}$$

**Table 1 | Comparison of the TDONN chip with state-of-the-art reconfigurable photonic processors**

| Technology | Throughout (TOPS)[a] | Computing density (TOPS/mm²)[b] | Energy efficiency (TOPS/W) | Task modality |
|---|---|---|---|---|
| MZI mesh[13] | 6.4 | N/A | N/A | Single |
| MZI mesh[18] | 14.4 | N/A | N/A | Single |
| MRR array[28] | 2.4 | 2.94 | N/A | Single |
| MRR array[38] | 0.8 | 6.12 | 1.18 | Single |
| Nanobeam array[58] | 4.8 | N/A | 1.32 | Single |
| PCM tensor core[34] | 28.8 | 10 | 3.33 | Single |
| PCM tensor core[35] | 180 | 1.48 | 2.16 | Single |
| This work | 217.6 | 447.7 | 7.28 | Multimodal |

[a]The throughput is estimated at a typical detection rate of 100 GHz.
[b]Regarding only the photonic chip.

where $T$ is the number of operations per second (OPS), $m$ is the number of hidden layers in the TDONN, $N^2$ is the effective matrix size, and $r$ is the detection rate of PDs. For the 1st to 4th hidden layer of TDONN chip, $m = 4$, $N = 16$, and for the 5th hidden layer, $m = 1$, $N^2 = 16 \times 4$, thus the total estimated throughput is 217.6 TOPS, which is two orders of magnitude higher than that of modern graphic processing units (GPUs), which typically perform at $10^{12}$ OPS[48]. The energy consumption of the TDONN chip mainly includes the chip itself and external components. For the TDONN chip, the modulation units and the on-chip diffractive units are realized by thermo-optical phase shifters, and the average energy consumption of each phase shifter is about 30 mW. Table 1 compares the key performance of our TDONN chip with other ONN architectures. The throughput of our TDONN chip is 217.6 TOPS, the computing density is 447.7 TOPS/mm², and the system-level energy efficiency is 7.28 TOPS/W, which has obvious advantages compared with other ONN architectures. Detailed estimation of key metrics and energy consumption information for external components are provided in Supplementary Note 11. The throughput of TDONN architecture mainly depends on the number and size of hidden layers, so the scalable TDONN architecture can be flexibly designed for high-throughput application scenarios. Due to the narrow width of the tunable diffractive units, massive diffractive units can be accommodated in one hidden layer, which helps to achieve high computing density. In addition, the TDONN chip only needs very low-power consumption to maintain or change the state of tunable diffractive units during the computation, so it can naturally be highly energy efficient while being reconfigurable. In contrast to bulky space diffraction systems and optical fiber systems, the key components of TDONN are integrated and can be fabricated on CMOS-compatible platforms, which is crucial for the commercialization of photonic computing chips.

Implementing nonlinear activation functions in the optical domain can avoid frequent optical-electrical conversions and improve the practicability of ONNs. Our recent work has successfully experimentally demonstrated two kinds of optical nonlinear function chips based on Ge/Si hybrid structure[49,50] and they can be integrated components in TDONN architecture to complement the lack of on-chip optical nonlinear activation. In addition, PCM-based devices have no static energy consumption, and require energy only when they switch to different states[34–36,51–53]. To further improve the energy efficiency of the TDONN chip, PCM is suitable to be used as trainable diffractive units to control the transmission of optical field by changing their states. For instance, the electrically programmable phase shifter based on PCM can replace the traditional thermal-optic phase shifter[51]. The electrical pulse for PCM configuration can be theoretically supplied by integrated circuits with standard CMOS technology. The control circuits are potential to be integrated on-chip level by photonic and microelectronic co-fabrication technology[54–57] to achieve higher I/O rate and higher integration.

In summary, we propose a TDONN architecture with in situ training capability and demonstrate its application in the multimodal inference with a 5-layer proof-of-concept chip. The training and inference of the chip are both performed in the optical domain, and the photonic-electronic prototype is controlled by a single FPGA framework without human intervention. Massive trainable photonic neurons in a compact footprint are key for the TDONN chip to support multimodal tasks. Compared with the state-of-the-art photonic processors, our TDONN chip has the advantages of high throughput (217.6 TOPS), high computing density (447.7 TOPS/mm²), high system-level computing efficiency (7.28 TOPS/W), low latency (30.2 ps), and strong reconfigurability. Moreover, we developed a chip-level drop-out mechanism to accelerate the training, achieving a 36.5% training acceleration. The chip has successfully implemented four-class classification in three different modalities (vision, audio, and touch) with an average accuracy of 85.7%, which is comparable to digital computers. Our work paves the way for the development of large-scale on-chip diffractive neural network with substantial versatility.

## Methods
### Device fabrication
The trainable diffractive optical neural network (TDONN) is fabricated on the silicon-on-insulator (SOI) platform with a 220-nm-thick silicon top layer and a 2-µm-thick buried oxide. We employ deep ultra-violet photolithography using a 248-nm stepper to define the waveguide patterns, followed by anisotropic dry etching of silicon. Subsequently, a thin layer of titanium nitride (TiN) is deposited as a resistive layer for the heaters, and a thin aluminum film is patterned as the electrical connection to the electrodes and heaters. Isolation trenches are created by etching the $SiO_2$ top cladding and Si substrate. The whole fabrication process is completed using CMOS-compatible processes.

### Experimental setup
The light source is an IDPHOTONICS CoBrite-DX laser source to provide tunable polarization-maintaining laser input. The polarization controller is used to maximize the coupling of the light source to the chip. The control of the TDONN chip is realized through an FPGA-based framework, where the integrated photonic and electronic devices can work together and achieve target function. The DAC module can be programmed to generate the required voltage, ranging from 0 to 10 V, up to 96 channels. The DAC module is used to drive the thermo-optic phase shifters of the TDONN chip, including the MZIs in the input layer and the diffractive units in the hidden layer. The input data is encoded through a DAC-driven MZI. PD array is used to convert optical waveforms into electrical signals, and each channel corresponds to a classification label. The ADC module samples the output voltage of each channel at 16-bit resolution. The cost function calculated based on the real-time sampling results is used to characterize the real-time state of the TDONN. The cost function is optimized by

updating the parameters of the diffractive units through automated feedback. Diffractive units that do not contribute to optimize CF will be deactivated to accelerate the convergence of training. The TDONN chip is attached to a TEC with temperature control accuracy of 0.01 °C to minimize the impact of environmental temperature fluctuations.

## Data availability

The data that support the findings of this study are available from the corresponding author upon request.

## Code availability

The codes that support the findings of this study are available from the corresponding author upon request.

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

## Acknowledgements
The authors acknowledge the financial support by the National Key Research and Development Program of China (2022YFB2804200), the National Natural Science Foundation of China (U21A20511).

## Author contributions
J.C. and J.D. conceived the idea. J.C. designed the silicon photonic chip. J.C. and J.L.Z. conducted the experiments and analyzed the data. B.W., W.Z., X.L., J.H.Z., Y.T., and H.Z. involved in the discussion and theoretical analysis. J.C. wrote the original manuscript with contributions from all co-authors. J.C., C.H., Q.Z., M.G., and J.D. revised and edited the manuscript. J.D. and X.Z. supervised and coordinated all the work.

## Competing interests
The authors declare no competing interests.
