## [Peer Review File · Nature Communications]

Multimodal deep learning using on-chip diffractive optics with in situ training capabilityReviewers' comments:

Reviewer #1 (Remarks to the Author):

The paper titled "Multimodal deep learning using on-chip diffractive optics with in situ training capability" proposed an optical computing architecture with integrated programmable diffractive optics. They basically reconfigured the on-chip TDONN by deploying the tunable heaters over the slab waveguide to manipulate the phase of diffracted light continuously. Based on this, three multimodal classification tasks were conducted with high accuracy. The manuscript is clearly written and well organized. Although the deploying of heaters to achieve on-chip light field control is not new, as seen in the literature (<https://onlinelibrary.wiley.com/doi/abs/10.1002/lpor.202300330>), the study about using large-scale heaters to configure on-chip TDONN has certain significance and novelty. In addition, the key structure of the chip in this work is similar to the design method published by the same group Cheng et al. (<https://doi.org/10.1021/acsphotonics.2c00716>) in the ACS Photonics journal last year. The main difference between the two works lies in the different ways in which the physical structure parameters of the chip are obtained. The previous work was based on inverse design to obtain parameters, while this work adopts gradient descent methods to obtain parameters under the premise that the physical structure of the chip is analyzed. However, there are not many statements and demonstrations regarding the physical structure analysis of the chip in this work. Before making my final decision about this work, I suggest the authors further address the following concerns.

1) The authors constructed a forward propagation model of TDONN using the Discrete Fourier Transform (DFT) method, an alternative expression for the diffraction integral to describe DONN. To prevent interference from reflected light, the model necessitates an open boundary within the diffractive region. However, according to the micrograph of the TDONN chip provided by the authors, the boundary is not open and it is more like an MMI whose physical description is quite different from DONN due to the strong light reflection. Consequently, the authors should verify the agreement between TDONN's theoretical calculations and its physical field simulation.

2) According to the manuscript, the in-situ training of TDONN appears to still rely on the classic error backpropagation algorithm, which necessitates an accurate physical model of TDONN for obtaining gradient information of the network. However, on line 214 of the manuscript, the authors mention that the crosstalk does exist in the TDONN due to thermal effect, and by configuring the parameters jointly can eliminate the impacts. Therefore, I have the following concerns about this training method:

a) It would be helpful for the readers if the authors provided information on the duration of the in-situ training required to complete a full task.

b) If the error backpropagation algorithm is applied, how to determine the gradient information that is disturbed by thermal crosstalk? How to propagate errors optically?

c) If another training algorithm that doesn't require gradient information is employed in this manuscript, such as particle swarm optimization or function learning, does that mean there is no need to construct an analytical model of TDONN? Please provide clarification.

3) Considering the substantial number of heaters deposited on the TDONN, thermal crosstalk cannot be overlooked even with the use of a TEC. It would be valuable for the authors to evaluate the stability of TDONN during in situ training and inference. I recommend including an experiment to validate the temperature stability during these processes and providing thermal simulations of the TDONN, both with and without using the proposed drop-out method.

Reviewer #2 (Remarks to the Author):

The manuscript by Cheng et al. offers an exploration into multimodal deep learning leveraging on-chip diffractive optics with in-situ training capabilities. The manuscript is well-organized and articulately written. However, certain aspects, the novelty and the metrics in particular, warrant further consideration.

-The proposed method bears resemblance to prior work in [1], which also explores thermal optical tuning for diffractive neural networks in photonic integrated circuits. Notably, [1] presents a model with 7 layers and four neurons in each layer, raising questions about the novelty of the current approach.

-The training methodology appears notably akin to the authors' previous work (ref.15), prompting concerns regarding the innovation quotient of this manuscript.

-The evaluation of TOPS (Tera Operations Per Second) seems somewhat misleading. For instance, despite claims of achieving 256 TOPS, the linear hidden layer with 16 inputs and four outputs only attains a speed of 6.4 TOPS at 100 G frequency. This discrepancy necessitates clarification.

-The utilization of a 100 G frequency, although purportedly based on experimental results, may be subject to limitations imposed by the bandwidth of thermal optical tuning. Moreover, the larger size of the 100G modulator raises questions about the validity of computing density evaluations, which ought to consider this factor.

Additional Comments:

-The task of classifying the number of fingers appears relatively straightforward, potentially warranting exploration of more challenging tasks to better demonstrate the model's capabilities.

-The inclusion of dropout regularization seems somewhat incongruent with the modest scale of the network, consisting of only 80 weights. Further justification for its incorporation would enhance clarity.

-The efficiency metric of 88TOPS/W, as elucidated in the supplementary information, requires clarification to differentiate between system-level efficiency and other metrics. A suggestion is made to include system-level efficiency in the main table for improved clarity.

-It would be valuable to incorporate a comparison between in-situ and in-silico training speeds in Figure 4, offering insights into the relative efficiency of these approaches.

[1] Chen, Tao, et al. "Programmable Parallel Optical Logic Gates on a Multimode Waveguide Engine." *Photonics*. Vol. 9. No. 10. MDPI, 2022.

Reviewer #3 (Remarks to the Author):

The authors proposed a trainable diffractive optical neural network (TDONN) chip that enables multimodal deep learning tasks. A 5-hidden-layer TDONN chip is fabricated and successfully implemented four-class classification in different modalities (vision, audio, and touch) and achieved 85.7% accuracy. In addition, the gradient descent algorithm and the drop-out mechanism realize the in situ training and fast convergence of photonic neurons in the optical domain. The experimental results are convincing and of high quality. Overall, the authors have demonstrated an impressive work that provides a new avenue for multimodal deep learning, which is currently very useful in artificial intelligence generated content. I recommend that this manuscript can be accepted for publication in *Nature Communications* after the authors address the following concerns.

1. Novelty:

a. Compared with other diffractive optical architectures, such as *Science* (361, p.1004, 2018), *Nature* (623, p.48-57, 2023), and *Nature Communications* (14, p.70, 2023), the key advantage of this work is the in situ training capability, while others are all fixed optical networks and can only perform single-modal tasks. What is the benefit brought by the in-situ training capability and how does it contribute to multimodal tasks?

b. Did the authors use compensation algorithms like *Nature Communications* (14, p.70, 2023)? In fact, additional digital post-processing can lead to frequent O/E conversion, which may have a significant impact on computing speed.

c. Mainstream photonic computing architectures based on MZI and MRR can implement a single linear photonic computation layer, such as *Nature Communications* (14, p.66, 2023), *Nature Communications* (13, p.7970, 2023), and *Nature Communications* (12, p.457, 2021). In these architectures, one computation instruction can be performed per light propagation, namely matrix-vector multiplication (MVM). However, the TDONN architecture proposed by the authors is very different from the architectures for MVMs. Instead of performing MVM calculation instructions, it implements an inference task. To facilitate readers to understand, it is suggested to add the comparison diagram of task-inference photonic processors and instruction-computing photonic processors.

2. Performance:

a. The description of performance is not comprehensive enough. Although potential throughput is an important indicator of a photonic processor, it cannot fully reflect the performance of a photonic processor in performing actual deep learning tasks. Another important indicator is the classification time. In particular, the training time is also an important indicator for the TDONN chip with in situ training capability. In practical optical-electronic computing system, the bottleneck may be the digital processing, so I suggest the authors provide the overall training and classification time in supplementary materials.

b. The energy consumption of the laser source is not properly discussed. In the energy consumption estimation, the power consumption of the laser source is only 0.04W, which is much lower than the actual power consumption of the laser. For instance, the pump temperature controller needs to be taken into account in the power consumption. If a benchtop laser is used in the experiments, the authors should evaluate it based on the actual or rated power consumption of the instrument to better fit the actual situation.

3. Scalability:

a. How does the TDONN architecture scale to more photonic neurons and layers? Will the energy efficiency of optical-electronic computing system grow linearly?

b. To expand the function of optical neural networks, the authors need to consider how to realize nonlinear activation in the future. How can nonlinear activation be incorporated into the proposed TDONN architecture?

4. Other questions and comments:

a. Some important works in the field of optical computing have been reported in recent months, such as Nature Photonics (17, p.1080-1088, 2023) and Optica (11, p.190-196, 2024), but these latest works are not included in the comparison table. I suggest the authors add these latest works to the comparison table.

b. In the feature extraction process of multi-modal data, the images of MNIST dataset are clipped (see Fig. 3a). Please explain the reason for the clipping, because it does not seem necessary.

Response letter

Dear Sir/Madam,

Thank you very much for taking time out of your busy schedule to review our manuscript entitled “Multimodal deep learning using on-chip diffractive optics with *in situ* training capability” (ID: NCOMMS-24-06613). We sincerely thank all reviewers for their highly constructive reviews and valuable feedback to improve the quality of our manuscript. We have modified the manuscript in accordance with their comments and suggestions. Here, we present a point-by-point reply (in blue) to the reviewers' comments, as well as the corresponding modifications in our main manuscript and supplementary materials (in red).

Best regards,

Prof. Jianji Dong^{1,2,*}

¹Wuhan National Laboratory for Optoelectronics, Huazhong University of Science and Technology, Wuhan 430074, China

²Optics Valley Laboratory, Wuhan 430074, China

*Corresponding author: jjdong@mail.hust.edu.cn

Point-by-Point Responses

Reply to Reviewer 1

General comment: The paper titled “Multimodal deep learning using on-chip diffractive optics with *in situ* training capability” proposed an optical computing architecture with integrated programmable diffractive optics. They basically reconfigured the on-chip TDONN by deploying the tunable heaters over the slab waveguide to manipulate the phase of diffracted light continuously. Based on this, three multimodal classification tasks were conducted with high accuracy. The manuscript is clearly written and well organized. Although the deploying of heaters to achieve on-chip light field control is not new, as seen in the literature (<https://onlinelibrary.wiley.com/doi/abs/10.1002/lpor.202300330>), the study about

using large-scale heaters to configure on-chip TDONN has certain significance and novelty. In addition, the key structure of the chip in this work is similar to the design method published by the same group Cheng et al. (<https://doi.org/10.1021/acsp Photonics.2c00716>) in the ACS Photonics journal last year. The main difference between the two works lies in the different ways in which the physical structure parameters of the chip are obtained. The previous work was based on inverse design to obtain parameters, while this work adopts gradient descent methods to obtain parameters under the premise that the physical structure of the chip is analyzed. However, there are not many statements and demonstrations regarding the physical structure analysis of the chip in this work. Before making my final decision about this work, I suggest the authors further address the following concerns.

Reply: First, we would like to thank the reviewer for constructive comments and suggestions. **We think that TDONN is obviously innovative compared with the previous work of our research group.** The TDONN chip integrates an input layer and five hidden layers, enabling multimodal classification tasks through training. This marks a significant difference from our previous work, and the training difficulty of the TDONN chip is noticeably higher than before. To train TDONN efficiently, we specially develop a drop-out mechanism to accelerate convergence. Furthermore, the previous chip structure is relatively simple (only matrix vector multiplication), and the problem of error accumulation for optical analog computation in running deep multilayer models has not yet emerged under such chip scale. **In summary, the innovativeness of this work can be concluded in the following three aspects:**

(1) Distinct computing paradigm. As shown in Fig. S1, the TDONN chip is a photonic processor based on task inference, while our previous work is a photonic processor based on matrix multiplication, and their computing paradigms are substantially different. Our previous work merely introduced the concept of on-chip training and conducted a preliminary verification on a small-scale photonic chip. For optical computing applications, we realized the matrix configuration and performed matrix-vector multiplication. In contrast, the TDONN chip simulates the functionality of a complete neural network and treats the entire chip as a 'black box'. Once training

is complete, it can directly execute inference tasks without the need for frequent matrix multiplication operations, marking a notable difference between the two.

Fig. S1. Two different computing paradigms. **a** Photonic processor based on matrix multiplication. **b** Photonic processor based on task inference.

(2) Innovative chip design. The TDONN chip integrates an input layer, and the amplitude modulation of input signal is implemented on the photonic chip. Moreover, the TDONN chip has five hidden layers, with 80 on-chip diffractive units in the hidden layers. The trainable parameters far exceed our previous work, necessitating the development of more efficient training methods. More trainable parameters enable the optical neural networks greater flexibility in learning and adapting to data. Especially for multimodal tasks, different tasks may require distinct parameter configuration. Optical neural networks with more adjustable parameters can adapt to various tasks by adjusting these parameters, facilitating multimodal learning.

(3) The first proposed drop-out mechanism for photonic neurons. During the training process of large-scale optical neural networks, the efficiency of iterative cycles can be improved by deactivating neurons that do not contribute to the target function. This accelerates the convergence of the training process in large-scale photonic computing chips. Additionally, the drop-out mechanism reduces neuronal dependencies. In neural networks, neurons may develop complex co-adaptations, potentially leading to overly complex models that are challenging to generalize. Drop-out breaks these co-

adaptations by randomly shutting down some neurons, simplifying the model and facilitating generalization.

Due to these three innovative aspects, the TDONN chip can support *in situ* training of massive optical parameters and accomplish multimodal inference tasks.

According to the reviewer's comments, we carefully revised our manuscript and the detailed point-by-point responses are listed below.

Comment 1: The authors constructed a forward propagation model of TDONN using the Discrete Fourier Transform (DFT) method, an alternative expression for the diffraction integral to describe DONN. To prevent interference from reflected light, the model necessitates an open boundary within the diffractive region. However, according to the micrograph of the TDONN chip provided by the authors, the boundary is not open and it is more like an MMI whose physical description is quite different from DONN due to the strong light reflection. Consequently, the authors should verify the agreement between TDONN's theoretical calculations and its physical field simulation.

Reply: Thanks for the professional comment. **Our theoretical model is established under ideal conditions and aims to be applicable to general diffractive architectures. Therefore, we do not presuppose that the boundary is open or not.** The hidden layers of this model contain numerous tunable diffractive units. During the propagation of light in this model, each diffractive unit introduces a phase shift, and the phase shift will contribute to the modulation of optical signals. This is the basic principle and key concept of this model. In our proof-of-concept chip, phase shift is introduced by applying voltages on resistive heaters to change the temperature of the waveguide and cladding materials, thereby changing the effective refractive index.

To visualize the effect of the voltage applied to the heater on the temperature, we simulate the thermal field of a single heater and multiple heaters when different voltages are applied. The thermal field simulation of a single heater is shown in Fig. S3a, and the thermal field simulation of multiple heaters is shown in Fig. S3b. **To quantify the effect of applied voltages on the output of the TDONN chip, we perform further simulation and experiment.** As shown in Fig. S3c, the monitor set at output port O₃ records the optical power when different voltages are applied to the

heater. We test the effect of the on-chip diffractive unit on the output of TDONN by simulation and experiment respectively, and the results are shown in Fig. S3d. **Simulation and experimental results show that the on-chip diffractive unit can effectively control the light field in TDONN, and verify agreement between TDONN's theoretical calculations and its physical field simulation.** It should be noted that we do not limit the method of introducing phase shift in our ideal theoretical model, rather, we treat phase shift as a tunable variable. Real experiments are indeed more complex than ideal conditions. As the reviewer mentioned, there is reflection of light at the boundary in practice. However, the entire diffractive region of the TDONN chip is considered as a 'black box' with numerous tunable diffractive units, and we do not pay specific attention to the details of light reflection. Experimental results show that light reflection does not affect the training effect of the TDONN, and it is feasible to train an optical neural network with multimodal inference capability. The trained neuron parameters may have minor deviations from those calculated by an ideal theoretical simulation model, but this does not affect the target function of the chip.

Fig. S3. Simulation of the TDONN. **a** Thermal field simulation of a single heater. **b** Thermal field simulation of multiple heaters. **c** Monitor the power of output port O_3 when tuning one heater. **d** Experiment and simulation results of output port O_3 .

Changes made in this revision:

(Manuscript page 3) “Phase shift is introduced by applying voltages on resistive heaters to effectively control the light propagation in TDONN (see **Supplementary Note S5** for detailed information).”

(Supplementary Note S5) “The hidden layers of this model contain numerous tunable diffractive units. During the propagation of light in this model, each diffractive unit introduces a phase shift, and the phase shift will contribute to the modulation of optical signals. This is the basic principle and key concept of this model. In our proof-of-concept chip, phase shift is introduced by applying voltages on resistive heaters to change the temperature of the waveguide and cladding materials, thereby changing the effective refractive index. To visually show the effect of the voltage applied to the heater on the temperature, we simulate the thermal field of a single heater and multiple heaters when different voltages are applied. The thermal field simulation of a single heater is shown in Fig. S3a, and the thermal field simulation of multiple heaters is shown in Fig. S3b. To quantify the effect of applied voltages on the output of the TDONN chip, we perform further simulation and experiment. As shown in Fig. S3c, the monitor set at output port O₃ records the optical power when different voltages are applied to the heater. We test the effect of the on-chip diffractive unit on the output of TDONN by simulation and experiment respectively, and the results are shown in Fig. S3d. Simulation and experimental results show that the on-chip diffractive unit can effectively control the light field in TDONN, and verify agreement between TDONN’s theoretical calculations and its physical field simulation.”

Comment 2: According to the manuscript, the in-situ training of TDONN appears to still rely on the classic error backpropagation algorithm, which necessitates an accurate physical model of TDONN for obtaining gradient information of the network. However, on line 214 of the manuscript, the authors mention that the crosstalk does exist in the TDONN due to thermal effect, and by configuring the parameters jointly can eliminate the impacts. Therefore, I have the following concerns about this training method:

Comment 2a: It would be helpful for the readers if the authors provided information on the duration of the in-situ training required to complete a full task.

Reply: Thanks for the constructive comment. We provide the information on the

duration of the *in situ* training required to complete a full task in **Supplementary Note S8**. Table S1 shows the estimated latency of the prototype and provides the training and inference time required to complete a full task. The latency of the prototype system mainly consists of four parts: light propagation, response time of TiN heater, DAC, and ADC. The latency of light propagation is only 30.2 ps per iteration, and the operating frequency of the prototype is limited by the response time of TiN heater. To match the response time of thermo-optic modulation, the system operating frequency of the prototype is set to 10 kHz in multimodal task demonstration. The training process for completing a full task involves approximately 1000 iterations, and after training is completed, inference can be achieved with just one forward propagation. More importantly, the state of the heaters in the diffractive region of the TDONN chip need not be changed after the training is completed, so high-speed electro-optic modulators and PDs can be used for multimodal inference tasks to significantly reduce the latency.

Table S1. Estimated latency of the TDONN prototype system.

Parameters of the prototype system	
Latency of light propagation	30.2 ps per iteration
Response time of TiN heater	10 μ s per iteration
Latency of the DAC	0.1 ms per iteration
Latency of the ADC	0.1 ms per iteration
Latency of training	
Number of iterations required	1000
Total time	0.21 s
Latency of inference	
Number of iterations required	1
Total time	0.21 ms

Changes made in this revision:

(Manuscript page 7) “The optical latency of the TDONN chip is approximately 30.2 ps (see **Supplementary Note S8** for detailed information).”

(Supplementary Note 8) “Table S1 shows the estimated latency of the prototype and provides the training and inference time required to complete a full task. The latency of the prototype system mainly consists of four parts: light propagation, response time of TiN heater, DAC, and ADC. The latency of light propagation is only 30.2 ps per iteration, and the operating frequency of the prototype is limited by the response time

of TiN heater. To match the response time of thermo-optic modulation, the system operating frequency of the prototype is set to 10 kHz in multimodal task demonstration. The training process for completing a full task involves approximately 1000 iterations, and after training is completed, inference can be achieved with just one forward propagation. More importantly, the state of the heaters in the diffractive region of the TDONN chip need not be changed after the training is completed, so high-speed electro-optic modulators and PDs can be used for multimodal inference tasks to significantly reduce the latency.”

Comment 2b: If the error backpropagation algorithm is applied, how to determine the gradient information that is disturbed by thermal crosstalk? How to propagate errors optically?

Reply: Thanks for the reviewer’s comment. Our theoretical model is established based on ideal conditions, aiming to provide the fundamental principles of the general TDONN architecture through analytical formulas. Since our analytical model is idealized, and thermal modulation is not the only modulation method for implementing the TDONN architecture, the gradient information of thermal crosstalk is not considered in the error backpropagation algorithm. However, in real experiments, **to compensate for thermal crosstalk during training, we treat the diffractive region of the TDONN chip as a "black box" with many tunable diffractive elements, and the influence of thermal crosstalk is also included.** By adjusting optical parameters of the entire "black box", the chip can ultimately achieve the target function. We set a cost function (CF) to calculate the gradient of network parameters and update the parameters in the opposite direction of the gradient, thereby gradually reducing the loss during training and improving the performance of the optical neural network. **In experiments, optical inference results are detected by a PD array, then pass through the ADC module and enter the FPGA, and the error information is obtained by comparing the optical inference results with theoretical values.** The calculation of the CF is performed by external circuits, and control signals are applied to the on-chip diffractive units through the DAC module to adjust the phase parameters

in the TDONN chip to perform specific feedback operations. Since changes in optical phase parameters affect the transmission of the optical field, the target of updating parameters in the opposite direction of the gradient can be achieved. The advantage of this approach is that it eliminates the need for time-consuming and complex electromagnetic simulations, and there is no need to set additional monitoring ports in the network. Instead, the chip function configuration for multimodal inference can be achieved through real-time training of optical parameters without human intervention.

Comment 2c: If another training algorithm that doesn't require gradient information is employed in this manuscript, such as particle swarm optimization or function learning, does that mean there is no need to construct an analytical model of TDONN? Please provide clarification.

Reply: Thanks for the reviewer's comment. **We agree with the reviewer's viewpoint that if an alternative training algorithm that does not require gradient information,** such as particle swarm optimization or function learning, is adopted in this paper, it is not necessary to construct an analytical model for TDONN. However, we think that the analytical model is helpful to explain and understand the optical transmission process of TDONN, which lays a foundation for subsequent research on this architecture. **Our research group previously proposed an efficient configuration scheme for MZI networks based on the particle swarm algorithm in Ref. [1],** which does not require the construction of an analytical model for the MZI network. The particle swarm algorithm is an optimization algorithm rooted in swarm intelligence. It simulates the foraging behavior of birds, representing each possible solution as a "particle" and searching for the optimal solution in the search space. In each iteration, particles update their velocity and position according to specific formulas, followed by a re-evaluation of their fitness values. This process continues until a termination condition is met, such as reaching the maximum number of iterations or finding an optimal solution that satisfies the precision requirements. The details of gradient information and light propagation are not involved in the training process of particle swarm algorithm.

In summary, the necessity of constructing an analytical model depends on the characteristics of the optical computing architecture and the training algorithm.

Nonetheless, constructing an analytical model can help readers understand the physical processes involved in the architecture.

Reference:

[1] Wu B., et al. Real-Valued Optical Matrix Computing with Simplified MZI Mesh. *Intelligent Computing* 2, 0047 (2023).

Comment 3: Considering the substantial number of heaters deposited on the TDONN, thermal crosstalk cannot be overlooked even with the use of a TEC. It would be valuable for the authors to evaluate the stability of TDONN during in situ training and inference. I recommend including an experiment to validate the temperature stability during these processes and providing thermal simulations of the TDONN, both with and without using the proposed drop-out method.

Reply: Thanks for the reviewer's comment. **We have added an experiment to verify the temperature stability of TDONN during *in situ* training and inference, and the detailed information is provided in Supplementary Note S9.** The experiment is performed with and without the proposed drop-out method respectively, and the results are shown in Fig. S7. The drop-out method accelerates the convergence of the training. Due to the large area of the TDONN chip, there are 80 heaters in the diffractive region, and the training process usually includes thousands of iterations, it will take several months to perform thermal simulation of the whole training process, which is contrary to the advantages of our training in the optical domain. As an alternative, **we provide heatmaps of the real-time heater power of the training process in Movies S1-S3,** which can visually show the heater configuration during training.

Fig. S7. Temperature stability test of the TDONN chip in multimodal tasks. **a** image classification, **b** audio classification, **c** tactile classification.

Changes made in this revision:

(Manuscript page 7) “To validate the temperature stability of the TDONN, an experiment is performed with and without using the proposed drop-out method, and the results is provided in **Supplementary Note S9.**”

(Supplementary Note S9) “To verify the temperature stability of TDONN, an experiment is performed with and without the proposed drop-out method respectively, and the results are shown in Fig. S7. The blue line shows the results without the drop-out method, and the orange line shows the results with the drop-out method. It can be observed that the drop-out method helps to accelerate the training and achieve the target faster. The temperature stability test is divided into three stages. The first stage is to optimize the TDONN chip at an initial temperature of 20 °C. With the increase of iterations, the cost function (CF) gradually converges to the maximum value until the function of the device meets the design goals. The second stage is used to demonstrate the degradation of device performance caused by external temperature changes. We use the temperature control module to gradually increase the external temperature of the TDONN chip from 20 to 30 °C. During the process of changing the temperature, the performance of the device begins to deteriorate significantly, and the CF value rapidly decays to near 0. The third stage is used to demonstrate the ability and robustness of the TDONN chip against environmental disturbances. The output of the chip is significantly abnormal due to external temperature changes, and the optimization program is activated and restarted. After several iterations at an external temperature of 30 °C, the CF value converges to the maximum value once again, indicating that only a short optimization is required for the device function to meet the design goal even after a drastic change in external temperature.”

Reply to Reviewer 2

General comment: The manuscript by Cheng et al. offers an exploration into multimodal deep learning leveraging on-chip diffractive optics with in-situ training capabilities. The manuscript is well-organized and articulately written. However,

certain aspects, the novelty and the metrics in particular, warrant further consideration.

Reply: We are grateful to the reviewer for the constructive comments and are delighted that the reviewer is interested in our work. Those comments are all valuable and very helpful for improving our manuscript, and significantly guide our research. We are happy to address all the comments below. We thank the reviewer for stimulating those improvements.

Comment 1: The proposed method bears resemblance to prior work in [1], which also explores thermal optical tuning for diffractive neural networks in photonic integrated circuits. Notably, [1] presents a model with 7 layers and four neurons in each layer, raising questions about the novelty of the current approach.

[1] Chen, Tao, et al. "Programmable Parallel Optical Logic Gates on a Multimode Waveguide Engine." *Photonics*. Vol. 9. No. 10. MDPI, 2022.

Reply: Thanks for the reviewer's comment. We have carefully read the work mentioned by the reviewer, which **implements basic logic gate operations using the MMI structure but does not involve the concept of optical neural networks**. We think that our TDONN architecture differs significantly from this work, and the novelty of our work is reflected in three aspects: working principle, architectural design, and training algorithm.

(1) Working principle. Our TDONN chip can perform multimodal inference tasks with one forward light propagation, and there is no need to cascade multiple chips. Previous work [1] could only achieve basic logic gate operations. However, implementing inference tasks in artificial intelligence often requires the cascading of numerous logic gates, which may not be feasible due to the insert loss.

(2) Architectural design. TDONN represents a complete optical neural network architecture, including an input layer, 5 hidden layers, and an output layer. The input layer contains a series of intensity modulation units to encode input data. The amplitude modulation of the input signal is performed on-chip, rather than using discrete components. Furthermore, the TDONN chip has 16 tunable modulation units and 80 adjustable diffractive units. More adjustable parameters make optical neural networks more flexible to adapt to various application scenarios, which is important for

accomplishing multimodal tasks.

(3) Training algorithm. To address the challenges posed by the large scale of the TDONN chip and the numerous adjustable diffractive units, **we develop a tailored gradient descent algorithm.** Additionally, **we firstly proposed a drop-out mechanism specifically designed for optical neural networks.** The combination of these two innovations enables efficient online training of the TDONN chip and accelerates convergence by 36.5%. Significantly, our proposed training algorithm and drop-out mechanism have great potential for large-scale optical computing chips.

In summary, we contend that the TDONN architecture differs substantially from previous work [1] and exhibits notable novelty in terms of working principle, architectural design, and training algorithm.

Comment 2: The training methodology appears notably akin to the authors' previous work (ref.15), prompting concerns regarding the innovation quotient of this manuscript.

Reply: Thanks for the reviewer's comment. **We think that the training methodology of TDONN is obviously innovative compared with the authors' previous work.** The computing paradigm of TDONN is innovative compared with the authors' previous work, and the TDONN chip integrates an input layer and 5 hidden layers, enabling multimodal classification tasks through training. **These innovation of fundamental principle and chip architecture mark a significant difference from our previous work, and the training difficulty of the TDONN chip is noticeably higher than before.** To train TDONN efficiently, we have specially developed a drop-out mechanism to accelerate convergence. Furthermore, the previous chip structure is relatively simple, and under such a chip scale, issues with optically simulated computations running deep multi-layer models had not yet emerged. **The drop-out mechanism for photonic neurons is first proposed and applied in TDONN.** During the training process of large-scale optical neural networks, the efficiency of iterative cycles can be improved by deactivating neurons that do not contribute to the target function. This accelerates the convergence of the training process in large-scale photonic computing chips. Additionally, the drop-out mechanism reduces neuronal dependencies. In neural networks, neurons may develop complex co-adaptations,

potentially leading to overly complex models that are challenging to generalize. Drop-out breaks these co-adaptations by randomly shutting down some neurons, simplifying the model and facilitating generalization.

In addition to the innovation of the training methodology, the computing paradigm and chip design of TDONN are also significantly innovative compared to the authors' previous work.

(1) Distinct computing paradigm. As shown in Fig. S1, the TDONN chip is a photonic processor based on task inference, while our previous work is a photonic processor based on matrix multiplication, and their computing paradigms are substantially different. Our previous work merely introduced the concept of on-chip training and conducted a preliminary verification on a small-scale photonic chip. For optical computing applications, we realized the matrix configuration and performed matrix-vector multiplication. In contrast, the TDONN chip simulates the functionality of a complete neural network and treats the entire chip as a 'black box'. Once training is complete, it can directly execute inference tasks without the need for frequent matrix multiplication operations, marking a notable difference between the two.

Fig. S1. Two different computing paradigms. **a** Photonic processor based on matrix multiplication. **b** Photonic processor based on task inference.

(2) Innovative chip design. The TDONN chip integrates an input layer, and the amplitude modulation of input signal is completed on the photonic chip. Moreover, the

TDONN chip has five hidden layers, with 80 on-chip diffractive units in the hidden layers. The trainable parameters far exceed our previous work, necessitating the development of more efficient training methods. More trainable parameters enable the optical neural networks greater flexibility in learning and adapting to data. Especially for multimodal tasks, different tasks may require distinct parameter configuration. Optical neural networks with more adjustable parameters can adapt to various tasks by adjusting these parameters, facilitating multimodal learning.

Due to these three innovations, the TDONN chip can support *in situ* training of massive optical parameters and accomplish multimodal inference tasks.

Comment 3: The evaluation of TOPS (Tera Operations Per Second) seems somewhat misleading. For instance, despite claims of achieving 256 TOPS, the linear hidden layer with 16 inputs and four outputs only attains a speed of 6.4 TOPS at 100 G frequency. This discrepancy necessitates clarification.

Reply: Thanks for the reviewer's suggestion. According to our investigation, the formula proposed by Yichen Shen et al. in paper [1] has been widely used for estimating the throughput of optical computing among recent high-level papers [1-7]. To make a relatively fair comparison with other works in the field of optical computing, we employed a similar methodology as referenced in [1-5] to calculate the expected throughput budget for the integrated chip. It should be noted that our TDONN chip contains five hidden layers, and **all hidden layers cannot be simply equivalent to one hidden layer, which shares the same viewpoint as the spatial diffraction-based optical computing in Ref. [8]**. Table S2 illustrates the specific calculation process for the scale and computing power of each hidden layer in TDONN. Based on Shen's formula, we calculate the throughput for five hidden layers individually. The chip itself has 16 output ports, and we use 4 of them in our experiments. Therefore, the fifth hidden layer has two scenarios: one with 16 output ports and another with 4 output ports. According to the actual experimental situation, we choose the fifth hidden layer with 4 output ports. At a typical frequency of 100 GHz, the potential throughput of each layer can be calculated. Summing the potential throughput of all five hidden layers, the total potential throughput of the TDONN chip can be obtained, which is 217.6 TOPS.

Table S2. Potential throughput estimation of the TDONN chip.

Layers	Scale	Potential throughput
1 st layer	16×16	2×16×16×100 G=51.2 TOPS
2 nd layer	16×16	2×16×16×100 G=51.2 TOPS
3 rd layer	16×16	2×16×16×100 G=51.2 TOPS
4 th layer	16×16	2×16×16×100 G=51.2 TOPS
5 th layer with 16 outputs	16×16	2×16×16×100 G=51.2 TOPS
5 th layer with 4 outputs	16×4	2×16×4×100 G=12.8 TOPS
Total layers		Total potential throughput
Total 5 layers with 16 outputs		256 TOPS
Total 5 layers with 4 outputs		217.6 TOPS

References:

- [1] Shen Y., et al. Deep learning with coherent nanophotonic circuits. Nat. Photonics 11, 441-446 (2017).
- [2] Feldmann J., et al. Parallel convolutional processing using an integrated photonic tensor core. Nature 589, 52-58 (2021).
- [3] Xu X., et al. 11 TOPS photonic convolutional accelerator for optical neural networks. Nature 589, 44-51 (2021).
- [4] Zhou T., et al. Large-scale neuromorphic optoelectronic computing with a reconfigurable diffractive processing unit. Nat. Photonics 15, 367-373 (2021).
- [5] Fu T., et al. Photonic machine learning with on-chip diffractive optics. Nat. Commun. 14, 70 (2023).
- [6] Xu S., Wang J., Yi S., Zou W. High-order tensor flow processing using integrated photonic circuits. Nat. Commun. 13, 7970 (2022).
- [7] Bai B., et al. Microcomb-based integrated photonic processing unit. Nat. Commun. 14, 66 (2023).
- [8] Lin X., et al. All-optical machine learning using diffractive deep neural networks. Science 361, 1004 (2018).

Changes made in this revision:

(Manuscript page 7) “Detailed estimation of key metrics and energy consumption information for external components are provided in **Supplementary Note S10**.”

(Supplementary Note S10) “Based on Eq. (S-14), we calculate the potential throughput

for five hidden layers individually. The scale of the fifth hidden layer is 16×4 , and the scale of other hidden layers is 16×16 . At a typical frequency of 100 GHz, the potential throughput of each layer can be calculated. Summing the potential throughput of all five hidden layers, the total potential throughput of the TDONN chip can be obtained, which is 217.6 TOPS.”

Comment 4: The utilization of a 100 G frequency, although purportedly based on experimental results, may be subject to limitations imposed by the bandwidth of thermal optical tuning. Moreover, the larger size of the 100G modulator raises questions about the validity of computing density evaluations, which ought to consider this factor.

Reply: Thanks for the constructive comment. At present, we use the thermo-optic MZIs as input data loading for the proof of principle of TDONN. We expect to adopt high-speed electro-optic MZMs to modulate 16-channel input data, which can achieve 100 GHz data loading. The weight adjustment of on-chip diffractive units is still the thermo-optical phase shifter, but the weights of them are fixed once the training of the TDONN is completed, indicating that we do not need to modulate the phase shifters anymore and the operating rate of TDONN will not be affected. In the field of optical computing, the typical frequency of 100 GHz is a commonly used estimated value [1, 2]. Hence, we have also adopted this typical frequency for potential performance estimation. To ensure fairness, the performance indicators for each work in the comparison tables in our paper are calculated using the same typical frequency of 100 GHz. In addition, we acknowledge that if high-speed electro-optic MZMs are used, the chip size will be increased. However, since they are not integrated in TDONN for the time being, we use the footprint of the TDONN chip to calculate the computing density.

References:

[1] Shen Y., et al. Deep learning with coherent nanophotonic circuits. *Nat. Photonics* 11, 441-446 (2017).

[2] Fu T., et al. Photonic machine learning with on-chip diffractive optics. *Nat. Commun.* 14, 70 (2023).

Additional Comments:

Comment 5: The task of classifying the number of fingers appears relatively

straightforward, potentially warranting exploration of more challenging tasks to better demonstrate the model's capabilities.

Reply: Thanks for the constructive comment. We further extended the application of the TDONN chip to the classification of gestures 5-8. Since the meaning of gestures 5-8 does not solely depend on the number of fingers, this classification task poses a greater challenge. Experimental results show that the trained TDONN chip can successfully recognize gestures 5-8, and the classification results are presented in Fig. S5.

Fig. S5. Classification of gestures 5-8. **a** gesture 5, **b** gesture 6, **c** gesture 7, **d** gesture 8.

Changes made in this revision:

(Manuscript page 7) “To better demonstrate the TDONN chip's capability, we further use the TDONN chip to classify more gestures, and the results are provided in **Supplementary Note S6.**”

(Supplementary Note S7) “To better demonstrate the TDONN chip's capability, we further use the TDONN chip to classify gestures 5-8. Since the meaning of gestures 5-8 does not solely depend on the number of fingers, this classification task poses a greater challenge. Experimental results show that the trained TDONN chip can successfully recognize gestures 5-8, and the classification results are presented in Fig. S5.”

Comment 6: The inclusion of drop-out regularization seems somewhat incongruent with the modest scale of the network, consisting of only 80 weights. Further

justification for its incorporation would enhance clarity.

Reply: Thanks for the valuable comment. **Although we only have 80 on-chip diffractive units to configure the weights, the drop-out mechanism also plays an important role, and its advantage will be more obvious when the number of on-chip diffractive units is larger.** The drop-out mechanism proposed in this paper aims to achieve fast convergence of training by deactivating photonic neurons that do not contribute to the training process. In the case of the DTONN chip, if a micro-heater fails to contribute to the improvement of the cost function after multiple adjustments, it will be deactivated and not configured in subsequent training sessions. The figure below illustrates the process of gradually deactivating some photonic neurons as training progresses. Initially, all photonic neurons are in an activated state. However, as training progresses, some neurons begin to be deactivated until the final state is reached. The drop-out mechanism reduces the number of optical parameters that need adjustment in later iterations, thus improving training efficiency. In fact, in the proof-of-concept chips for optical computing, 80 weights are already considered a large number. Typically, several hundred or even thousands of iterations are required to achieve the target function in experiments, and the drop-out mechanism can make these iterations more efficient.

Figure. Schematic of the drop-out mechanism used for photonic neurons.

Comment 7: The efficiency metric of 88 TOPS/W, as elucidated in the supplementary information, requires clarification to differentiate between system-level efficiency and other metrics. A suggestion is made to include system-level efficiency in the main table

for improved clarity.

Reply: Thanks for the valuable suggestion. Since the energy efficiency in the optical computing field usually does not include the power consumption of digital back-end modules, the energy efficiency metric we initially used was calculated based on this criterion. System-level energy efficiency is a more objective metric to evaluate the optical computing system, and it includes the power consumption of the digital backend module. **We agree with the reviewer and use the system-level energy efficiency metric in the revised manuscript to improve clarity.** The total power consumption is 29.88 W, including laser source, TDONN chip, photodetectors, TEC, and digital backend modules. The potential throughput is 217.6 TOPS, so **the system-level efficiency can be calculated as 7.28 TOPS/W.** In addition, the energy consumption of each item is listed in Table S3 in **Supplementary Note S10.**

Table S3 Estimated power consumption of the TDONN.

Module	Components	Power (W)
Laser source	DFB laser	0.5
Data modulation	Heaters (Input=16)	$0.03 \times 16 = 0.48$
On-chip diffractive network	Heaters (N=80)	$0.03 \times 80 = 2.4$
Photodetectors	PD driver (Output=4)	$0.5 \times 4 = 2$
TEC	TEC for laser source	1.5
	TEC for TDONN chip	3
Digital backend	FPGA control circuits	20
External benchtop instruments	Arbitrary waveform generator	50
	Oscilloscope	120
Total power consumption (include benchtop instruments)		199.88
Total power consumption (exclude benchtop instruments)		29.88

Changes made in this revision:

(Manuscript page 2) “Our TDONN chip achieves a potential throughput of 217.6 tera-operations per second (TOPS) with high computing density (447.7 TOPS/mm²), high energy efficiency (7.28 TOPS/W)”

(Manuscript page 8) “The throughput of our TDONN chip is 217.6 TOPS, the computing density is 447.7 TOPS/mm², and the system-level energy efficiency is 7.28 TOPS/W, which has obvious advantages compared with other ONN architectures. Detailed estimation of key metrics and energy consumption information for external components are provided in **Supplementary Note S10.**”

(Manuscript page 9) “our TDONN chip has the advantages of high throughput (217.6 TOPS), high computing density (447.7 TOPS/mm²), high system-level computing efficiency (7.28 TOPS/W)”

(Supplementary Note S10) “The estimated total power consumption of the TDONN system is approximately 29.88 W, and the system-level energy efficiency of the TDONN can be calculated as 7.28 TOPS/W. Notably, a substantial portion of the power consumption stems from the digital backend. When the power consumption of the digital backend module is not considered, the energy efficiency is 22.02 TOPS/W. The energy efficiency of the TDONN can be significantly improved by scaling up the on-chip diffractive network.”

Comment 8: It would be valuable to incorporate a comparison between in-situ and in-silico training speeds in Figure 4, offering insights into the relative efficiency of these approaches.

Reply: Thanks for the valuable suggestion. We have added the time it takes the digital computer to train the optical neural network model in **Supplementary Note S8**. To evaluate the relative efficiency of optical training, we train the same multimodal classification model using a digital computer, and the time required for training is recorded. In the digital computer, it takes 1226.47 s to complete the training of the model. Under the operating frequency of 10 kHz, TDONN only needs 0.21 s to complete the training of the model, achieving more than 5800× acceleration.

Changes made in this revision:

(Supplementary Note S10) “To evaluate the relative efficiency of optical training, we train the same multimodal classification model using a digital computer (Intel(R)

Core(TM) i9-12900K CPU, 32 GB RAM), and the time required for training is recorded. In the digital computer, it takes 1226.47 s to complete the training of the model. Under the operating frequency of 10 kHz, TDONN only needs 0.21 s to complete the training of the model, achieving more than $5800\times$ acceleration.”

Reply to Reviewer 3

General comment: The authors proposed a trainable diffractive optical neural network (TDONN) chip that enables multimodal deep learning tasks. A 5-hidden-layer TDONN chip is fabricated and successfully implemented four-class classification in different modalities (vision, audio, and touch) and achieved 85.7% accuracy. In addition, the gradient descent algorithm and the drop-out mechanism realize the in situ training and fast convergence of photonic neurons in the optical domain. The experimental results are convincing and of high quality. Overall, the authors have demonstrated an impressive work that provides a new avenue for multimodal deep learning, which is currently very useful in artificial intelligence generated content. I recommend that this manuscript can be accepted for publication in Nature Communications after the authors address the following concerns.

Reply: We are grateful to the reviewer for the constructive comments and are delighted that the reviewer is interested in our work. Those comments are all valuable and very helpful for improving our manuscript, and we have carefully revised our manuscript.

1. Novelty:

Comment 1a: Compared with other diffractive optical architectures, such as Science (361, p.1004, 2018), Nature (623, p.48-57, 2023), and Nature Communications (14, p.70, 2023), the key advantage of this work is the *in situ* training capability, while others are all fixed optical networks and can only perform single-modal tasks. What is the benefit brought by the in-situ training capability and how does it contribute to multimodal tasks?

Reply: Thanks for the reviewer’s valuable comment. **As the reviewer pointed out,**

most fixed optical networks can only perform single-modal tasks. The key advantage of our TDONN work lies in its *in situ* training capability, which enables multimodal deep learning tasks. The benefits of the training capability can be summarized in the following three aspects:

First, broader application scenarios. Due to its stronger adaptability and performance, the TDONN chip with flexible parameter adjustment and *in situ* training capability can be applied to a wider range of scenarios and tasks. Whether it is image recognition, speech recognition, or tactile recognition, the chip function can be programmed according to specific requirements, and the performance of the chip can be optimized by adjusting on-chip optical parameters. The TDONN chip can achieve multimodal deep learning tasks through *in situ* training.

Second, the TDONN architecture supports continual learning. For the reconfigurable optical neural network, it can continuously learn and evolve through parameter training when new data or tasks arise. Although various optical networks represented by diffractive neural networks are emerging, the basic optical computing units in existing optical computing systems are generally limited by their solidified structures and low scalability. The TDONN architecture supports continual learning through training, representing a lifelong learning optical computing architecture.

Third, high fabrication precision and compensation algorithms in digital computers are not necessary. During chip fabrication, fabrication errors are inevitably introduced, leading to significant degradation in the accuracy of optical computing. For fixed optical neural networks, a common solution is to compensate fabrication errors through algorithms in digital computer, which undoubtedly causes additional delays and post-processing. However, the TDONN chip, with its *in situ* training capability, can compensate for deviations in fabrication by real-time adjustment of the parameters of the on-chip diffractive units. Meanwhile, the fabrication process of the TDONN chip is simple and available in commercial foundries. The chip fabrication requires no high-precision processing equipment and has a large processing tolerance.

Comment 1b: Did the authors use compensation algorithms like Nature Communications (14, p.70, 2023)? In fact, additional digital post-processing can lead to frequent O/E conversion, which may have a significant impact on computing speed.

Reply: Thanks for the reviewer's valuable comment. **We do not use compensation algorithms** like those described in Nature Communications (14, p.70, 2023). Fabrication error in photonic chip will significantly degrade the accuracy of optical computing. For non-trainable optical neural networks, a common solution is to compensate for these errors through algorithms in digital computers, which undoubtedly introduces additional delays and post-processing. However, the TDONN chip, equipped with online training capabilities, can adjust the parameters of the on-chip diffractive units in real-time to compensate for fabrication errors. As a result, the TDONN chip eliminates the need for extra digital post-processing and frequent O/E conversions. After completing the training process, the TDONN chip only requires a single forward propagation of light to realize multimodal optical inference.

Comment 1c: Mainstream photonic computing architectures based on MZI and MRR can implement a single linear photonic computation layer, such as Nature Communications (14, p.66, 2023), Nature Communications (13, p.7970, 2023), and Nature Communications (12, p.457, 2021). In these architectures, one computation instruction can be performed per light propagation, namely matrix-vector multiplication (MVM). However, the TDONN architecture proposed by the authors is very different from the architectures for MVMs. Instead of performing MVM calculation instructions, it implements an inference task. To facilitate readers to understand, it is suggested to add the comparison diagram of task-inference photonic processors and instruction-computing photonic processors.

Reply: Thanks for the reviewer's valuable comment. We have added a comparison diagram between the task inference photonic processor and the matrix multiplication photonic processor to facilitate readers' understanding of the novelty of our TDONN architecture. The matrix multiplication processor can only perform one matrix multiplication calculation per light propagation, requiring frequent O/E conversions. In

contrast, the task inference processor can accomplish one inference task per light propagation without the need for frequent O/E conversions, resulting in low latency and power consumption.

Fig. S1. Two computing paradigms. **a** Photonic processor based on matrix multiplication. **b** Photonic processor based on task inference.

Changes made in this revision:

(Manuscript page 2) “A comparison diagram between TDONN and the traditional matrix multiplication architecture is given in **Supplementary Note S1.**”

(Supplementary Note S1) “The photonic processor based on matrix multiplication can only perform one matrix multiplication calculation per light propagation, requiring frequent O/E conversions. In contrast, the photonic processor based on task inference can accomplish one inference task per light propagation without the need for frequent O/E conversions, resulting in low latency and power consumption.”

2. Performance:

Comment 2a: The description of performance is not comprehensive enough. Although potential throughput is an important indicator of a photonic processor, it cannot fully reflect the performance of a photonic processor in performing actual deep learning tasks. Another important indicator is the classification time. In particular, the training time is also an important indicator for the TDONN chip with in situ training capability. In

practical optical-electronic computing system, the bottleneck may be the digital processing, so I suggest the authors provide the overall training and classification time in supplementary materials.

Reply: Thanks for the reviewer’s valuable comment. After completing the fabrication and packaging of the TDONN chip, we further design a TDONN prototype by integrating modules such as FPGA, DAC, ADC, and develop a user graphical interface to demonstrate optical multimodal inference. **Table S1 shows the estimated latency of the prototype and provides the training and inference time required to complete a full task.**

Table S1. Estimated latency of the TDONN prototype system.

Parameters of the prototype system	
Latency of light propagation	30.2 ps per iteration
Response time of TiN heater	10 μ s per iteration
Latency of the DAC	0.1 ms per iteration
Latency of the ADC	0.1 ms per iteration
Latency of training	
Number of iterations required	1000
Total time	0.21 s
Latency of inference	
Number of iterations required	1
Total time	0.21 ms

The latency of the prototype system mainly consists of four parts: light propagation, response time of TiN heater, DAC, and ADC. The latency of light propagation is only 30.2 ps per iteration, and the operating frequency of the prototype is limited by the response time of TiN heater. To match the response time of thermo-optic modulation, the system operating frequency of the prototype is set to 10 kHz in multimodal task demonstration. The training process for completing a full task involves approximately 1000 iterations, and after the training is completed, inference can be achieved with just one forward propagation. More importantly, the state of the heaters in the diffractive region of the TDONN chip need not be changed after the training is completed, so high-speed electro-optic modulators and photodetectors can be used for multimodal inference tasks to significantly reduce the latency.

(Manuscript page 7) “The latency of the TDONN chip is approximately 30.2 ps (see **Supplementary Note S8** for detailed information).”

(Supplementary Note 8) “Table S1 shows the estimated latency of the prototype and provides the training and inference time required to complete a full task. The latency of the prototype system mainly consists of four parts: light propagation, response time of TiN heater, DAC, and ADC. The latency of light propagation is only 30.2 ps per iteration, and the operating frequency of the prototype is limited by the response time of TiN heater. To match the response time of thermo-optic modulation, the system operating frequency of the prototype is set to 10 kHz in multimodal task demonstration. The training process for completing a full task involves approximately 1000 iterations, and after training is completed, inference can be achieved with just one forward propagation. More importantly, the state of the heaters in the diffractive region of the TDONN chip need not be changed after the training is completed, so high-speed electro-optic modulators and PDs can be used for multimodal inference tasks to significantly reduce the latency.”

Comment 2b: The energy consumption of the laser source is not properly discussed. In the energy consumption estimation, the power consumption of the laser source is only 0.04W, which is much lower than the actual power consumption of the laser. For instance, the pump temperature controller needs to be taken into account in the power consumption. If a benchtop laser is used in the experiments, the authors should evaluate it based on the actual or rated power consumption of the instrument to better fit the actual situation.

Reply: Thanks for the reviewer’s valuable comment. Our energy consumption estimation is based on mainstream estimation methods in the field of optical computing, as references [1-3]. To make the power consumption estimation more aligned with actual power consumption conditions, **we agreed with the reviewer's suggestion and use the actual power consumption of the DFB laser to calculate the total power consumption and related performance metrics.** In addition, TEC is used for temperature control in the experiment, so we also include the power consumption of

TEC in the total power consumption. The power consumption information of all modules is given in Table S3 in **Supplementary Note S10**, where the power consumption of the DFB laser is 0.5 W and the power consumption of the TEC used for the laser source is 1.5 W.

Table S3 Estimated power consumption of the TDONN.

Module	Components	Power (W)
Laser source	DFB laser	0.5
Data modulation	Heaters (Input=16)	$0.03 \times 16 = 0.48$
On-chip diffractive network	Heaters (N=80)	$0.03 \times 80 = 2.4$
Photodetectors	PD driver (Output=4)	$0.5 \times 4 = 2$
TEC	TEC for laser source	1.5
	TEC for TDONN chip	3
Digital backend	FPGA control circuits	20
External benchtop instruments	Arbitrary waveform generator	50
	Oscilloscope	120
Total power consumption (include benchtop instruments)		199.88
Total power consumption (exclude benchtop instruments)		29.88

References:

- [1] Fu T., et al. Photonic machine learning with on-chip diffractive optics. Nat. Commun. 14, 70 (2023).
- [2] Xu S., Wang J., Yi S., Zou W. High-order tensor flow processing using integrated photonic circuits. Nat. Commun. 13, 7970 (2022).
- [3] Bai B., et al. Microcomb-based integrated photonic processing unit. Nat. Commun. 14, 66 (2023).

(Manuscript page 8) “The throughput of our TDONN chip is 217.6 TOPS, the computing density is 447.7 TOPS/mm², and the system-level energy efficiency is 7.28 TOPS/W, which has obvious advantages compared with other ONN architectures. Detailed estimation of key metrics and energy consumption information for external components are provided in **Supplementary Note S10**.”

(Supplementary Note S10) “The estimated total power consumption of the TDONN

system is approximately 29.88 W, and the system-level energy efficiency of the TDONN can be calculated as 7.28 TOPS/W. Notably, a substantial portion of the power consumption stems from the digital backend. When the power consumption of the digital backend module is not considered, the energy efficiency is 22.02 TOPS/W. The energy efficiency of the TDONN can be significantly improved by scaling up the on-chip diffractive network.”

3. Scalability:

Comment 3a: How does the TDONN architecture scale to more photonic neurons and layers? Will the energy efficiency of optical-electronic computing system grow linearly?

Reply: Thanks for the reviewer’s valuable comment. TDONN is a highly scalable multi-layer optical neural network architecture, where the number of hidden layers and the capacity of each layer can be easily expanded according to task requirements. Specifically, when designing the TDONN chip, more layers can be added in the diffractive region to increase the network's depth, and more on-chip diffractive units can be placed in each layer to increase the capacity of the hidden layers. Since the energy consumption of the photonic chip itself only accounts for a small portion of the system's total energy consumption, while the external circuits and benchtop instruments constitute most of the system's power consumption, scaling up the TDONN architecture does not result in a linear increase in energy efficiency.

Comment 3b: To expand the function of optical neural networks, the authors need to consider how to realize nonlinear activation in the future. How can nonlinear activation be incorporated into the proposed TDONN architecture?

Reply: Thanks for the reviewer’s comment. Implementing nonlinear activation functions in the optical domain can further improve the practicability of ONNs. Our recent work has successfully experimentally demonstrated two kinds of optical nonlinear function chips based on Ge/Si hybrid structure [1-2] and they can be integrated components in TDONN architecture to complement the lack of on-chip optical nonlinear activation. The nonlinear activation device based on Ge/Si hybrid

structure can be set at the output layer. When the optical signal output from the hidden layer enters the nonlinear activation device in the output layer, it will be selectively passed or suppressed according to whether it reaches the activation threshold, and then enter the PD array.

References:

- [1] Wu B., et al. Low-threshold all-optical nonlinear activation function based on a Ge/Si hybrid structure in a microring resonator. *Opt. Mater. Express* 12, 970-980 (2022).
- [2] Li H., et al. All-Optical Nonlinear Activation Function Based on Germanium Silicon Hybrid Asymmetric Coupler. *IEEE J. Sel. Top. Quantum Electron.* 29, 1-6 (2023).

4. Other questions and comments:

Comment 4a: Some important works in the field of optical computing have been reported in recent months, such as *Nature Photonics* (17, p.1080-1088, 2023) and *Optica* (11, p.190-196, 2024), but these latest works are not included in the comparison table. I suggest the authors add these latest works to the comparison table.

Reply: Thanks for the reviewer's comment. We have added these recent works mentioned by the reviewer to the comparison table in our revised manuscript.

Table 1 Comparison of the TDONN chip with state-of-the-art reconfigurable photonic processors.

Technology	Throughput (TOPS) ^a	Computing density (TOPS/mm ²) ^b	Energy efficiency (TOPS/W)	Task modality
MZI mesh ¹³	6.4	N/A	N/A	Single
MZI mesh ¹⁸	14.4	N/A	N/A	Single
MRR array ²⁸	2.4	2.94	N/A	Single
MRR array ³⁸	0.8	6.12	1.18	Single
Nanobeam array ⁴⁹	4.8	N/A	1.32	Single
PCM tensor core ³⁴	28.8	10	3.33	Single
PCM tensor core ³⁵	180	1.48	2.16	Single
This work	217.6	447.7	7.28	Multimodal

^aThe throughput is estimated at a typical detection rate of 100 GHz.

^bRegarding only the photonic chip.

Comment 4b: In the feature extraction process of multi-modal data, the images of

MNIST dataset are clipped (see Fig. 3a). Please explain the reason for the clipping, because it does not seem necessary.

Reply: Thanks for the reviewer's comment. In the MNIST dataset, handwritten digits are not always positioned at the center of the image. Without clipping the image to center the digits, the fully connected layers may disproportionately focus on irrelevant information at the edges of the image, leading to model overfitting. By clipping, we can ensure that the digits are centered in the image, allowing the model to concentrate more on the features of the digits themselves and improving recognition accuracy. Additionally, handwritten digit images in the MNIST dataset may contain noise and background interference. Such interference information may affect the judgment of the model and needs to be removed by clipping. Since the input image only retains the core region of the digit itself, the accuracy of the model can be improved.

REVIEWER COMMENTS

Reviewer #1 (Remarks to the Author):

This paper is reviewed before and has been rejected to publication according the Editor's decision. This revised version still meets the main problems – novelty and modeling. Also, the author's reply to comments are not satisfactory. The reviewer believes that to thoroughly explain this issue, the manuscript needs to be reconsidered and significantly revised.

1. Novelty: In the response letter and Supplementary Notes 1, the authors assert that the proposed paradigm differs from photonic matrix multiplication. The reviewer disagree with the authors' perspective. Although the proposed TDONN can be treated as a 'black box,' it is important to note that the TDONN does not introduce any nonlinear activations in the optical domain. Therefore, its essence still lies in matrix multiplications, making it no different from other photonic matrix multiplication schemes, and it requires frequent O/E conversion when processing feature vectors during inference tasks.

2. Modeling and error backpropagation: The authors claim that the TDONN functions as a black box, which does not consider boundaries when building the propagation model. If this is the case, there is no explicit gradient in the training process because there is no analytical model to describe it. Therefore, where does the gradient information come from when applying error backpropagation? The authors need to clarify this point. Additionally, in Supplementary Notes 3, the error propagation is based on the no-boundary diffraction model described in Supplementary Notes 2. However, the TDONN is stated to be a black box with no describable model, which presents a clear contradiction. The authors must address how its error propagation can be described.

3. The major issue is the author's selective disregard of reflections at the MMI boundary, leading to the failure of the basic physical process modeling. As previously suggested by the reviewer, the boundaries will reflect some light, which will reach the second heating electrode. Therefore, considering only the diffraction between layers is insufficient and does not represent the ideal conditions claimed by the author. Consequently, the reviewer believes that the modeling of the diffraction process is disconnected from the structure, and the author's "black box" modeling of this structure precisely demonstrates the unnecessary nature of the diffraction analysis. Hence, the reviewer finds a significant logical flaw in the manuscript. The reviewer suggests eliminating the diffraction analysis of this structure altogether and directly treating the structure as a "black box" in the design.

4. For the thermal crosstalk, the reviewer's comments focus on the long stability for TDONN when it carries tasks. But the response only gives the heater configurations during training. This process only shows the tunability of the heaters instead of stability.

Reviewer #2 (Remarks to the Author):

The authors have fully addressed my concerns. Both the results and concepts are well presented to demonstrate its potential for future applications.

Reviewer #3 (Remarks to the Author):

I appreciate the authors response. My concerns have been addressed and I support the publication of this work on Nature Communications.

Response letter

Dear Sir/Madam,

Thank you very much for taking time out of your busy schedule to review our manuscript entitled “Multimodal deep learning using on-chip diffractive optics with *in situ* training capability” (ID: NCOMMS-24-06613). We sincerely thank all reviewers for their highly constructive reviews and valuable feedback to improve the quality of our manuscript. We have modified the manuscript in accordance with their comments and suggestions. Here, we present a point-by-point reply (in blue) to the reviewers' comments, as well as the corresponding modifications in our main manuscript and supplementary materials (in red).

Best regards,

Prof. Jianji Dong^{1,2,*}

¹Wuhan National Laboratory for Optoelectronics, Huazhong University of Science and Technology, Wuhan 430074, China

²Optics Valley Laboratory, Wuhan 430074, China

*Corresponding author: jjdong@mail.hust.edu.cn

Point-by-Point Responses

Reply to Reviewer 1

General comment: This paper is reviewed before and has been rejected to publication according the Editor’s decision. This revised version still meets the main problems – novelty and modeling. Also, the author’s reply to comments are not satisfactory. The reviewer believes that to thoroughly explain this issue, the manuscript needs to be reconsidered and significantly revised.

Reply: We are grateful to the reviewer for the constructive comments. We have carefully read your comments and are committed to addressing them to improve the quality of our manuscript. We will revise our manuscript based on the feedback of the reviewer and provide point-by-point responses to ensure that each comment is properly

addressed. Regarding the two problems mentioned by the reviewer: novelty and modeling, **we add a detailed introduction in the revised version to the novelty of the TDONN architecture to explain its unique and valuable contribution** to the field of optical computing in Supplementary Note 1. In terms of modeling, we acknowledge that there are some discrepancies in details between our proposed ideal propagation model and actual experiments, such as the errors and waveguide boundaries mentioned by the reviewer. **We have added explanations of the ideal propagation model in Supplementary Note 2 and 3 to avoid misunderstandings.** In addition, **we have added a long-term stability test of TDONN** based on the reviewer's comment to verify the long-term stability of the TDONN prototype. We believe that based on these modifications, the quality of the manuscript will be greatly improved to meet the high publication standards of *Nature Communications*. Thanks again for the reviewer's time and expertise. According to the reviewer's comments, we have carefully revised our manuscript and the point-by-point responses are listed below.

Comment 1: Novelty: In the response letter and Supplementary Notes 1, the authors assert that the proposed paradigm differs from photonic matrix multiplication. The reviewer disagree with the authors' perspective. Although the proposed TDONN can be treated as a 'black box,' it is important to note that the TDONN does not introduce any nonlinear activations in the optical domain. Therefore, its essence still lies in matrix multiplications, making it no different from other photonic matrix multiplication schemes, and it requires frequent O/E conversion when processing feature vectors during inference tasks.

Reply: Thanks for the reviewer's comment. We agree with part of the reviewer's comment that TDONN does not introduce nonlinear activation in the optical domain, so it is essentially a linear transformation from the input space to the output space. However, the computing paradigm of TDONN is significantly different from traditional matrix multiplication processors. The workflows of these two types of photonic processors are shown in Fig. S2. **Fig. S2a illustrates the workflow of traditional matrix multiplication processors, where the photonic chip is treated as a matrix**

kernel, and each light propagation can perform a matrix multiplication operation. **This scheme represents precise computation**, corresponding to the calculations in electrical neural networks. Since a neural network model contains numerous matrix multiplications, it necessitates multiple loading operations of new input data and the utilization of the photonic processor to execute matrix multiplication operations. This computing paradigm poses a challenge: every matrix multiplication involves O/E conversion during the computation process. Additionally, the output data from the photonic processor is stored in electrical cache and need to be reshaped into the final inference result after all computations are completed. **In summary, the traditional matrix multiplication architectures have two notable drawbacks: (1) frequent O/E conversions and (2) the need for electrical data storage and data reshape during computations.**

Fig. S2. The workflows of different photonic processors. **a** Photonic processor based on matrix multiplication. **b** Photonic processor based on task inference.

The TDONN architecture is a photonic processor architecture based on task-specific inference, and its workflow is shown in Fig. S2b. In this scheme, the photonic chip can be regarded as a "black box" functionally equivalent to an on-chip linear neural network, where each light propagation can perform an inference task. **Unlike precise computation, this scheme is a form of fuzzy computation** that can directly map the input space (feature vectors) to the output space (classification labels) by adjusting the trainable units on the photonic chip. By comparing the contents in the dashed boxes of the two types of workflows in Fig. S2, it can be seen that the TDONN architecture

differs significantly from other photonic matrix multiplication schemes. Specifically, **the TDONN architecture requires only one forward propagation of light and two O/E conversions during the inference process**, and it eliminates the need for electrical data storage and reshape, which can effectively overcome the shortcomings of traditional matrix multiplication architectures. Therefore, we believe that our TDONN architecture represents a novel optical computing architecture with significant innovations compared to other optical computing schemes.

In addition, **recent research demonstrates that increasing the number of phase modulation layers in diffractive neural networks can enhance their expressive capabilities [1]**. Aydogan Ozcan et al. also experimentally demonstrated that deeper diffractive networks with larger numbers of trainable surfaces can cover a higher-dimensional subspace of the complex-valued linear transformations. Compared to single-layer models, multi-layer models possess depth advantages in classification tasks [2]. **As a multi-layer diffractive network architecture, TDONN exhibits stronger expressive and generalization abilities compared to traditional single-layer architectures based on matrix multiplication**, naturally endowed with the potential to perform multimodal deep learning tasks.

Last but not least, **our TDONN architecture has the potential to integrate optical nonlinear activation**. Previous work of our research team has demonstrated that the Ge/Si hybrid structure can achieve nonlinear activation [3-5]. Our next research task is to develop a TDONN chip with optical nonlinear activation. In fact, we have made preliminary progress. The following Fig. 1 shows a TDONN chip with optical nonlinear activation. The Ge/Si hybrid structure can be designed between diffractive layers or at the output region to achieve on-chip nonlinear activation. Currently, we are still optimizing relevant fabrication processes and structural parameters of this chip. The current progress indicates that our TDONN architecture has the potential to integrate on-chip optical nonlinear activation.

Fig. 1. TDONN chip with Ge/Si-based optical nonlinear activation.

References:

- [1] Zheng M., Liu W., Shi L., Zi J. Diffractive neural networks with improved expressive power for gray-scale image classification. *Photonics Res.* **12**, 1159-1166 (2024).
- [2] Kulce O., Mengu D., Rivenson Y., Ozcan A. All-optical information-processing capacity of diffractive surfaces. *Light-Sci. Appl.* **10**, 25 (2021).
- [3] Shi Y., *et al.* Nonlinear germanium-silicon photodiode for activation and monitoring in photonic neuromorphic networks. *Nat. Commun.* **13**, 6048 (2022).
- [4] Wu B., Li H., Tong W., Dong J., Zhang X. Low-threshold all-optical nonlinear activation function based on a Ge/Si hybrid structure in a microring resonator. *Opt. Mater. Express* **12**, 970-980 (2022).
- [5] Li H., Wu B., Tong W., Dong J., Zhang X. All-Optical Nonlinear Activation Function Based on Germanium Silicon Hybrid Asymmetric Coupler. *IEEE J. Sel. Top. Quantum Electron.* **29**, 1-6 (2023).

Changes made in this revision:

(Supplementary Note 1) “The workflows of these two types of photonic processors are shown in Fig. S2. Fig. S2a illustrates the workflow of traditional matrix multiplication processors, where the photonic chip is treated as a matrix kernel, and each light propagation can perform a matrix multiplication operation. This scheme represents precise computation, corresponding to the calculations in electrical neural networks. The computational process of deep neural network requires repeated invocation of the matrix multiplication processor, which is essentially a time-division multiplexing of

matrix multiplication, resulting in significant delays. Additionally, frequent O/E conversions and digital-to-analog conversions will also lead to extra power consumption. Since a neural network model contains numerous matrix multiplications, it necessitates multiple loading operations of new input data and the utilization of the photonic processor to execute matrix multiplication operations. This computing paradigm poses a challenge: every matrix multiplication involves O/E conversion during the computation process. Another problem is data storage and recovery. In conventional architectures, the output data from the photonic processor is usually stored in electrical cache and need to be reshaped into the final inference result after all computations are completed.”

(Supplementary Note 1) “On the contrary, our TDONN architecture is a photonic processor architecture based on task-specific inference, and its workflow is shown in Fig. S2b. In this scheme, the photonic chip can be regarded as a "black box" functionally equivalent to an on-chip linear neural network, where each light propagation can perform an inference task. Unlike precise computation, this scheme is a form of fuzzy computation that can directly map the input space (feature vectors) to the output space (classification labels) by adjusting the trainable units on the photonic chip. By comparing the contents in the dashed boxes of the two types of workflows in Fig. S2, it can be seen that the TDONN architecture differs significantly from other photonic matrix multiplication schemes. Specifically, the TDONN architecture requires only one forward propagation of light and two O/E conversions during the inference process, and it eliminates the need for electrical data storage and reshape, which can effectively overcome the shortcomings of traditional matrix multiplication architectures.”

Comment 2: Modeling and error backpropagation: The authors claim that the TDONN functions as a black box, which does not consider boundaries when building the propagation model. If this is the case, there is no explicit gradient in the training process because there is no analytical model to describe it. Therefore, where does the gradient information come from when applying error backpropagation? The authors need to clarify this point. Additionally, in Supplementary Notes 3, the error propagation is

based on the no-boundary diffraction model described in Supplementary Notes 2. However, the TDONN is stated to be a black box with no describable model, which presents a clear contradiction. The authors must address how its error propagation can be described.

Reply: Thanks for the constructive comment. In Supplementary Note 2 and 3, we provide a general theoretical model of diffractive neural networks that can be used for the propagation process analysis of spatial and integrated schemes. It should be mentioned that **the theoretical model is intended to help broad readers understand the propagation of light in the diffractive neural network.** Although the propagation model is not an accurate model, researchers working on diffractive neural network usually use it as a theoretical guidance to show the internal physics of diffractive chip [6-8]. When training this theoretical model in a digital computer, the gradient information of the backpropagation can be obtained. **However, the conditions of the actual experiments are not ideal, and the training of photonic chips does not strictly follow this theoretical model.** In the training process of the actual experiment, the explicit gradient cannot be directly obtained, and we treat TDONN as a 'black box' with numerous trainable weight parameters. Since there is no explicit gradient, we update the weight parameters of the diffractive network by detecting the optical response of the output layer in real time. Specifically, in the TDONN prototype, the output of the photonic chip is detected by the PD array in real time, and the real-time cost function is calculated as the evaluation index in the digital back-end. Then, the voltage applied to the on-chip diffractive unit is configured by the FPGA-based control framework to improve the cost function value, and finally the target function is realized. **To avoid misunderstandings, we carefully revised the manuscript and related Supplementary Notes, and we added detailed explanations of the differences between actual experiments and theoretical model.**

References:

- [6] Fu T., *et al.* Photonic machine learning with on-chip diffractive optics. *Nat. Commun.* **14**, 70 (2023).
- [7] Huang Y., Fu T., Huang H., Yang S., Chen H. Sophisticated deep learning with on-

chip optical diffractive tensor processing. *Photonics Res.* **11**, 1125-1138 (2023).

[8] Cheng Y., *et al.* Photonic neuromorphic architecture for tens-of-task lifelong learning. *Light-Sci. Appl.* **13**, 56 (2024).

Changes made in this revision:

(Manuscript page 2) “To help understand the physics of diffraction in the TDONN architecture, we build a general theoretical model of diffractive neural networks to describe the light forward propagation and the error backward propagation, and the details can be found in **Supplementary Note 2** and **Supplementary Note 3**.”

(Manuscript page 2) “In actual experiments, the waveguide boundary will reflect some light, which may reach the next diffractive layer, thus the training of photonic chip does not strictly follow the theoretical model. To effectively optimize the diffraction network in experiments, we treat TDONN as a 'black box' with numerous trainable weight parameters, and update the weight parameters of the diffractive network by detecting the optical response of the output layer in real time.”

(Supplementary Note 3) “When training this theoretical model in a digital computer, the gradient information of the backpropagation can be obtained. However, the conditions of the actual experiments are not ideal, and the training of photonic chip does not strictly follow this theoretical model. In the training process of the actual experiment, the explicit gradient cannot be directly obtained, and we treat TDONN as a 'black box' with numerous trainable weight parameters. Since there is no explicit gradient, we update the weight parameters of the diffractive network by detecting the optical response of the output layer in real time. Specifically, in the TDONN prototype, the output of the photonic chip is detected by the PD array in real time, and the real-time cost function value is calculated as the evaluation index in the digital back-end. Then, the voltage applied to the on-chip diffractive unit is configured by the FPGA-based control framework to improve the cost function value, and finally the target function is realized.”

Comment 3: The major issue is the author's selective disregard of reflections at the MMI boundary, leading to the failure of the basic physical process modeling. As

previously suggested by the reviewer, the boundaries will reflect some light, which will reach the second heating electrode. Therefore, considering only the diffraction between layers is insufficient and does not represent the ideal conditions claimed by the author. Consequently, the reviewer believes that the modeling of the diffraction process is disconnected from the structure, and the author's "black box" modeling of this structure precisely demonstrates the unnecessary nature of the diffraction analysis. Hence, the reviewer finds a significant logical flaw in the manuscript. The reviewer suggests eliminating the diffraction analysis of this structure altogether and directly treating the structure as a "black box" in the design.

Reply: Thanks for the reviewer's comment. **In our TDONN chip, the width of the waveguide in the diffraction region is relatively large, reaching 350 μm , thus the light reflected by the boundary is relatively little, and its impact on the diffraction process is relatively limited.** That is the reason why we previously analyzed it under ideal conditions. The target of modeling the diffraction process is to provide a general theoretical model of light propagation, which can be constructed and trained on a digital computer to help broad readers understand the propagation process. However, due to factors such as waveguide boundaries, **the actual situation does not strictly conform to the theoretical model, and we have added relevant explanations in Supplementary Note 2 and 3.** In practical experiments, to efficiently train the photonic chip, we consider TDONN as a "black box" and use intelligent algorithms to adjust the on-chip trainable units to directly achieve the mapping from input features to output inference labels. **We agree with the reviewer's comment that there are some differences between the modeling of the diffraction process and the actual structure.** To address this logical flaw, we have removed the analysis of the diffraction process from the main text and directly treated the structure as a "black box" in the design according to the reviewer's suggestion. In the supplementary materials, we retain the diffraction analysis model in Supplementary Note 2 and 3 to help readers understand the propagation of light in diffractive neural networks. Meanwhile, we have added explanations of the differences between the theoretical model and actual experiments in the diffraction analysis model to avoid misunderstandings.

Changes made in this revision:

(Manuscript page 2) “To help understand the physics of diffraction in the TDONN architecture, we build a general theoretical model of diffractive neural networks to describe the light forward propagation and the error backward propagation, and the details can be found in **Supplementary Note 2** and **Supplementary Note 3**.”

(Manuscript page 2) “In actual experiments, the waveguide boundary will reflect some light, which may reach the next diffractive layer, thus the training of photonic chip does not strictly follow the theoretical model. To effectively optimize the diffraction network in experiments, we treat TDONN as a 'black box' with numerous trainable weight parameters. Since there is no explicit gradient, we update the weight parameters of the diffractive network by detecting the optical response of the output layer in real time.”

(Supplementary Note 2) “It should be mentioned that this theoretical model is intended to help broad readers understand the propagation of light in the diffractive neural network. However, the conditions of the actual experiments are not ideal, for example, the waveguide boundary will reflect some light, which will reach the next diffractive layer, thus the actual light propagation does not strictly follow this theoretical model.”

Comment 4: For the thermal crosstalk, the reviewer’s comments focus on the long stability for TDONN when it carries tasks. But the response only gives the heater configurations during training. This process only shows the tunability of the heaters instead of stability.

Reply: Thank you for the reviewer's constructive comments. **We have added a new experiment to verify the long-term stability of TDONN when performing tasks, with detailed information provided in Supplementary Note 10.** The experiment consists of two stages: training and stability test. During the training stage, we train the TDONN chip to an optimal state using intelligent algorithms. In the stability test stage, we maintain the voltage applied to the on-chip trainable units and keep the trained TDONN prototype to operate for more than 6 hours. Throughout the experiment, we record the changes in the cost function over time, and the experimental results are shown in Fig. S9. Since the time of stability test is much longer than the training time,

to simultaneously show both the training and stability testing in the figure, we record the CF values in real-time during the training stage and once per second during the stability testing stage. Thanks to the TEC temperature control system and FPGA-based control framework, the TDONN prototype can maintain its original optimal state and successfully perform the target task even after 6 hours of completion of training. **The experimental results indicate that the TDONN prototype exhibits excellent long-term stability.**

Fig. S9. Long-term stability test of the TDONN chip.

Changes made in this revision:

(Manuscript page 8) “In addition, the results of long-term stability test of the TDONN are provided in **Supplementary Note 10.**”

(Supplementary Note 10) “To verify the long-term stability of TDONN when performing classification tasks, an experiment is performed, and the results are shown in Fig. S9. The experiment consists of two stages: training and stability test. During the training stage, we train the TDONN chip to an optimal state using intelligent algorithms. In the stability test stage, we maintain the voltage applied to the on-chip trainable units and keep the trained TDONN prototype to operate for more than 6 hours. Throughout the experiment, we record the changes in the cost function over time. Since the time of stability test is much longer than the training time, to simultaneously show both the training and stability testing in the figure, we record the CF values in real-time during the training stage and once per second during the stability testing stage. Thanks to the

TEC temperature control system and FPGA-based control framework, the TDONN prototype can maintain its original optimal state and successfully perform the target task even after 6 hours of completion of training. The experimental results indicate that the TDONN prototype exhibits excellent long-term stability.”

Reply to Reviewer 2

General comment: The authors have fully addressed my concerns. Both the results and concepts are well presented to demonstrate its potential for future applications.

Reply: Thank you for your thorough evaluation of our manuscript and for providing such positive feedback. We are committed to further exploring the possibilities of our findings and hope to contribute significantly to the advancement of academic community.

Reply to Reviewer 3

General comment: I appreciate the authors response. My concerns have been addressed and I support the publication of this work on Nature Communications.

Reply: Thank you for your positive feedback and support for our manuscript. We believe that our research has the potential to make a significant contribution to the field of optical computing, and we are excited about the opportunity to share our findings with a wider audience through publication on Nature Communications.

REVIEWERS' COMMENTS

Reviewer #1 (Remarks to the Author):

The reviewer thanks again the authors for the response to the comments. Despite the authors responses, the reviewer still believe that the overall scientific contribution is limited and not matching the standard of the journal.

Response letter

Dear Sir/Madam,

Thank you very much for taking time out of your busy schedule to review our manuscript entitled “Multimodal deep learning using on-chip diffractive optics with *in situ* training capability” (ID: NCOMMS-24-06613). We sincerely thank all reviewers for their highly constructive reviews and valuable feedback to improve the quality of our manuscript. We have modified the manuscript in accordance with their comments and suggestions. Here, we present a point-by-point reply (in blue) to the reviewers' comments.

Best regards,

Prof. Jianji Dong^{1,2,*}

¹Wuhan National Laboratory for Optoelectronics, Huazhong University of Science and Technology, Wuhan 430074, China

²Optics Valley Laboratory, Wuhan 430074, China

*Corresponding author: jjdong@mail.hust.edu.cn

Point-by-Point Responses

Reply to Reviewer 1

General comment: The reviewer thanks again the authors for the response to the comments. Despite the authors responses, the reviewer still believe that the overall scientific contribution is limited and not matching the standard of the journal.

Reply: Thank you very much for taking the time to review our manuscript and for providing your valuable feedback. We have addressed the concerns of the reviewer, and the quality of this manuscript has been significantly improved. We believe that our research has the potential to make an important contribution to the field of optical computing, and we remain open to any further suggestions or feedback.